

# Ice-shelf damming in the glacial Arctic Ocean: dynamical regimes of a basin-covering kilometre thick ice shelf

Johan Nilsson[1,2], Martin Jakobsson[3,2], Chris Borstad[4], Nina Kirchner[5,2], Göran Björk[6], Raymond T. Pierrehumbert[7], and Christian Stranne[3,2]

[1]Department of Meteorology, Stockholm University
[2]Bolin Centre for Climate Research, Stockholm University
[3]Department of Geological Sciences, Stockholm University
[4]The University Centre in Svalbard
[5]Department of Physical Geography, Stockholm University
[6]Department of Marine Sciences, University of Gothenburg
[7]Department of Physics, University of Oxford

*Correspondence to:* J. Nilsson (nilsson@misu.su.se)

**Abstract.** Recent geological and geophysical data suggest that a one-kilometre thick ice shelf extended over the glacial Arctic Ocean during Marine Isotope Stage 6, about 140 000 years ago. Here, we theoretically analyse the development and equilibrium features of such an ice shelf, using scaling analyses and a one-dimensional ice-sheet–ice-shelf model. We find that the dynamically most consistent scenario is an ice shelf with a nearly uniform thickness that covers the entire Arctic Ocean. Further, the ice shelf have two regions with distinctly different dynamics: a vast interior region covering the central Arctic Ocean and an exit region towards the Fram Strait. In the interior region, which is effectively dammed by the Fram Strait constriction, there are strong back stresses and the mean ice-shelf thickness is controlled primarily by the horizontally-integrated mass balance. A narrow transitions zone is found near the continental grounding line, in which the ice-shelf thickness decreases offshore and approaches the mean basin thickness. If the surface accumulation and mass flow from the continental ice masses are sufficiently large, the ice-shelf thickness grows to the point where the ice shelf grounds on the Lomonosov Ridge. As this occurs, the back stress increases in the Amerasian Basin and the ice-shelf thickness becomes larger there than in the Eurasian Basin towards the Fram Strait. Using a one-dimensional ice-dynamic model, the stability of equilibrium ice-shelf configurations without and with grounding on the Lomonosov Ridge are examined. We find that the grounded ice-shelf configuration should be stable if the two Lomonosov Ridge grounding lines are located on the opposites sides of the ridge crest, implying that the downstream grounding line is located on a downward sloping bed. This result shares similarities with the classical result on marine ice-sheet stability of Weertman, but due to interactions between the Amerasian and Eurasian ice-shelf segments the mass flux at the downstream grounding line decreases rather than increases with ice thickness.

## 1 Introduction

Based on analogies with conditions in present-day West Antartica, Mercer (1970) proposed that the Arctic Ocean during full glacial conditions should have been covered by kilometre thick ice shelves. Building on this notion and ice dynamical



considerations, Hughes et al. (1977) postulated that an Arctic ice-sheet–ice-shelf complex formed during the Last Glacial Maximum (LGM) between about 19,000 and 26,500 years ago (19 to 26,5 ka). A key motivation for their hypothesis was that the glacial marine ice sheets around the Arctic Ocean must have been stabilised by buttressing from a thick basin-covering ice shelf (Weertman, 1974). Over the years, the hypothesis of a basin-covering glacial Arctic Ocean ice shelf has been refined

and expanded in various aspects (Lindstrom and MacAyeal, 1986; Hughes, 1987; Grosswald and Hughes, 1999), but it has remained controversial. In fact, many subsequent studies that have mapped ice erosion on ridges and bathymetric highs in the central Arctic Ocean have attributed these traces to grounding of ice shelves that extended from the continental margins of northern Greenland, North America, and the Chukchi Borderland, but did not cover the entire the Arctic Basin (Jakobsson, 1999; Polyak et al., 2001; Jakobsson et al., 2010; Dowdeswell et al., 2010; Niessen et al., 2013; Jakobsson et al., 2014).

New data obtained during the icebreaker expedition with *Oden* in 2014 motivated Jakobsson et al. (2016) to revise the extent of glacial Arctic Ocean ice shelves: Previously unmapped ridges and bathymetric highs in the deep Arctic Ocean proved to be marked by scouring ice shelves and remapping of areas where ice erosion had been identified revealed flow directions that, when brought together with other existing data, pointed to a coherent basin-scale ice shelf. At the absolute minimum, the ice shelf must have covered all of the Amerasian Basin and have been grounded on the Lomonosov Ridge. However, ice erosion

documented north of the Fram Strait led Jakobsson et al. (2016) to propose that the ice shelf covered the entire Arctic Ocean (Fig. 1).

The purpose of this work is to reconsider the earlier Arctic Ocean ice-shelf scenarios (Mercer, 1970; Hughes et al., 1977; Grosswald and Hughes, 1999) in light of new data (Jakobsson et al., 2016) and new insights in ice-sheet–ice-shelf dynamics (Schoof and Hewitt, 2013). We will here use a one-dimensional ice-sheet–ice-shelf model and scaling analyses to examine

some key dynamical questions from a theoretical perspective. To begin with, there is an inception problem: how could shelf-ice expand laterally over the entire Arctic Basin? A next logical question concerns the evolution of ice-shelf thickness, controlled by net accumulation and dynamical thinning due to ice flow. Furthermore, a fully-developed kilometre thick ice shelf will ground on the Lomonosov Ridge, which yields an ice-shelf complex with three grounding lines: Transitions between grounded and floating ice are encountered at the upstream continental margin and on each side of the Lomonosov Ridge (see Figs.

1 and 5). This configuration introduces some novel dynamics related to interactions between floating and grounded ice: A fundamental question is whether such an ice-shelf–ice-sheet system is stable (Weertman, 1974; Schoof, 2012).

In addition, the data poses an intriguing puzzle in a broader glacial context: Several of the ice-grounding events in the deep Arctic Ocean have been dated based on the stratigraphic units resting directly above the ice-eroded surfaces, which indicate that the oldest undisturbed sediments were deposited during the Marine Isotope Stage 5 between 130 to 80 ka (Jakobsson et al.,

2010, 2016). Theses analyses place the traces of a kilometre thick Arctic shelf ice to the penultimate glaciation during the Marine Isotope Stage 6 (MIS 6) some 140 ka, rather than to the LGM. Remarkably, this suggests that the shelf ice formed in the Arctic Ocean during the LGM never grew thick enough to erode the bathymetric highs in the central basin.

Studies of the large Antarctic Filchner-Ronne and Ross ice shelves provide important information on the dynamics of glacial Arctic Ocean ice shelves, and relevant results will be reviewed here. However, the glacial Arctic problem has intriguing features

that sets it apart. At its full extent, the Arctic Ocean ice-shelf area was nearly an order of magnitude larger than that of the Ross





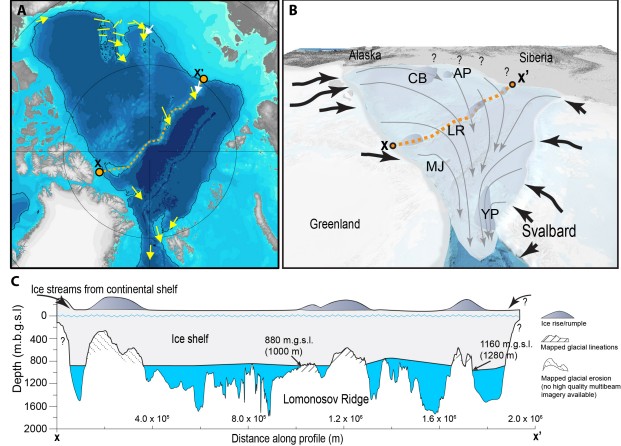

**Figure 1.** Outline of the glacial Arctic Ocean ice shelf adopted from Jakobsson et al. (2016). Panel a) shows present-day land distribution and water depth (blue colours) and observationally-inferred ice-shelf flow directions (arrows, see the text for details). The black line shows 1000 m isobath. The Lomonosov Ridge (along the dotted line) separates the Amerasian Basin from the Eurasian Basin, located towards the Fram Strait between Greenland and Svalbard. Panel b) depicts qualitatively the ice-shelf extent and flow based on ice-erosion data from the Chukchi Border Land (CB), the Arlis Plateau (AP), the Lomonosov Ridge (LR), the Morris-Jessup Rise (MR), and the Yermak Plateau. Question marks indicate possible East Siberian ice sheets (Niessen et al., 2013; Jakobsson et al., 2014). Panel c) shows a section along the Lomonosov Ridge, illustrating how the shelf grounds on ridge and the sub-ice channels that allow water circulation into the Amerasian ice cavity.

Ice Shelf. Further, an Arctic ice shelf extending over the entire central basin will not have a wide open margin as it would meet the relatively narrow Fram Strait, which constricts ice export from the Arctic Ocean. This geometrical setting provides a strong damming effect on the ice shelf. Some recent studies have examined dynamical aspects of glacial ice shelves in the Arctic Ocean (Jakobsson et al., 2005; Colleoni et al., 2010; Kirchner et al., 2013; Colleoni et al., 2016a). In particular, Colleoni et al.

5  (2016a) used a numerical ice-sheet–ice-shelf model to examine possible configurations of glacial East Siberian ice shelves. Notably, in a few of their simulations, which were run for 30 000 years, nearly 2 km thick shelf ice eventually extended over the entire Arctic Ocean. However, Colleoni et al. (2016a) focused on the local ice-shelf features off East Siberia and did not specifically analyse the simulations that yielded basin-covering Arctic Ocean ice shelves.

In the present study, we will theoretically analyse the dynamics of a basin-covering Arctic Ocean ice shelf. Our aim is to

10  identify leading-order dynamical balances, which provide general quantitative insights of how a horizontally extensive and strongly embayed ice shelf operates and interacts with contiguous ice sheets. We will in particular examine

1. how the interplay between net accumulation, dynamical thinning, and buttressing affects the evolution and equilibrium thickness of an Arctic Ocean ice shelf;

2. how grounding on the Lomonosov Ridge affects the structure and stability of the ice-sheet–ice-shelf complex.



Below, we introduce the geographical setting of the glacial Arctic Ocean ice shelf and discuss qualitatively ice-shelf inception scenarios. Thereafter, we proceed with the theoretical analyses (sections 3 and 4) and conclude by discussing analogies with Antarctic ice shelves and to consider some glacial implications (section 5).

## 2  Background

### 2.1  Geographical setting of an Arctic Ocean ice shelf under fully-glaciated conditions

Figure 1 summarises the geographical setting and outlines the Arctic Ocean ice-shelf distribution at fully-glaciated MIS 6 conditions synthesised by Jakobsson et al. (2016). Flow directions inferred from mapped sea-floor ice erosion across the deep Arctic Ocean point to a coherent ice-shelf flow directed towards the Fram Strait, located between Greenland and Svalbard (Fig. 1a,b). On the Lomonosov Ridge, which divides the Amerasian and Eurasian sub-basins, traces left by moving ice are found as deep as 1280 m below the present sea level (Fig. 1c). Accounting for a lower glacial sea level of 120 m, Archimedes' principle gives an ice-shelf thickness of nearly 1300 m at this grounding point. Ice groundings reaching more that 1000 m below the present-day sea level are also found on the Arlis Plateau off the eastern Siberian Continental Shelf (Niessen et al., 2013; Jakobsson et al., 2016). Along the East Siberian continental margin, there are few locations where the inferred ice-flow directions vary with water depth and between neighbouring sites. This erosion pattern may indicate ice flow associated with an East Siberian ice-sheet (Niessen et al., 2013), which extent can have varied during the MIS 6 (Svendsen et al., 2004; Colleoni et al., 2016b). During some stage of the MIS 6, a large and active East Siberian ice sheet may have fluxed grounded ice towards the central Arctic Basin (white arrows in Fig. 1a)[1]; whereas the ice-sheet configuration during other stages have yielded an ice flow preferentially directed towards the Fram Strait (yellow arrows in Fig. 1a). However, despite that the ice flow from the continental margins presumably evolved over time, the Arctic Ocean ice-shelf flow depicted in Fig. 1b should be representative of the fully-glaciated MIS 6 conditions.

Dynamical arguments, to be further presented in section 3, and numerical ice-shelf modelling (Lindstrom and MacAyeal, 1986; Colleoni et al., 2016a), suggest that the Arctic Ocean ice shelf should have two regions characterised by different features (Fig. 2): An inner strongly embayed central Arctic region with fairly uniform ice-shelf thickness and an exit region located towards the Fram Strait, which terminates at a calving front in the Nordic Seas. When the ice shelf grounded on the Lomonosov Ridge, the ice thickness should have become greater in the Amerasian Basin than in the Eurasian Basin. The ice cavity that formed beneath the Arctic Ocean ice shelf was vast and extensive, with a mean water depth exceeding 2 km even during fully glaciated conditions. The water exchange between the Eurasian and Amerasian parts of the ice cavity was then restricted to a few deep channels crossing the Lomonosov Ridge (Fig. 1c).

As a reference for the theoretical analyses of ice-shelf flow developed in section 3, it is instructive to here supply a few numbers characterising the Arctic Ocean ice shelf at full glacial conditions. Notably, the interior ice-shelf area is about $5 \cdot 10^6$ km$^2$, corresponding to that of the deep Central Arctic Ocean beyond the shelf breaks (Jakobsson, 2002). This is nearly one

---

[1]See also Fig. 7 in Jakobsson et al. (2014), which synthesises observations from the East Siberian continental margin prior to the *Oden* 2014 expedition





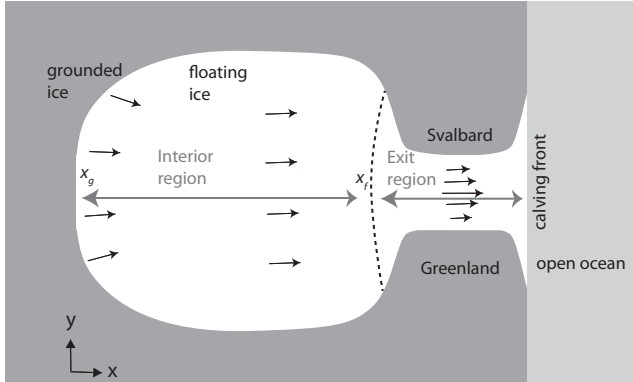

**Figure 2.** Outline of the two dynamical zones in the Arctic Ocean ice shelf, divided by the dashed line. The interior "dammed" ice-shelf region, where the ice thickness is nearly uniform and controlled by the horizontally-integrated mass balance [Eqs. (19,21)], comprises the bulk of the ice-shelf area and is subjected to pronounced back stress from the Fram Strait constriction. The exit region in the Fram Strait area, which terminates at a calving front, has dynamical similarities with moderately embayed large Antarctic ice shelves. The black arrows indicate ice flow and the two double-headed arrows are along the $x$–$z$ transect shown in Fig. 3. In most of the exit region, there is an approximate balance between lateral shear stress and hydrostatic pressure gradient [Eq. (23)], but the leading order ice-shelf momentum balance changes closer to the calving front (Hindmarsh, 2012).

order of magnitude larger than the area of the present-day Ross Ice Shelf. Climate-model simulations have provided some constraints on ice mass balances in the glacial Arctic: the combined surface snow accumulation on the Arctic shelf and influx of continental ice to the Arctic Basin have been simulated to lay in the range 1000–3000 km$^3$ yr$^{-1}$ in liquid water equivalents (Bigg and Wadley, 2001; Colleoni et al., 2010, 2016a). This amounts to a positive mass-balance contribution of 0.2–0.6 m yr$^{-1}$ per unit shelf area. The basal oceanic melting is less constrained by modelling studies, but could be about 0.2 m yr$^{-1}$ (MacAyeal, 1984; Jakobsson et al., 2016). For comparison, basal melt rates of the Filchner and East Ross ice shelves are estimated to be about 0.4 m yr$^{-1}$ (Rignot et al., 2013). Taking 1000 km$^3$ yr$^{-1}$ as a representative value for the net ice-shelf accumulation, one can infer the velocity in a 1000 m thick ice shelf: over the nearly 2000 km wide Lomonosov Ridge, ice velocities are expected to be a few hundred meters per year. In the narrower Fram Strait, which deep part is about 200 km wide, ice velocities are expected to reach a few kilometres per year.

## 2.2  Ice-shelf inception scenarios

Transitions from interglacial to glacial climates occur when the Earth's orbital variations cause anomalously low summer insolation in the Northern Hemisphere (e.g. Pierrehumbert, 2012). This reduces summer ablation and promotes accumulation of snow and formation of glaciers and ice sheets in the high northern latitudes. Growing circumpolar continental ice sheets will partly flow polewards, discharging ice into floating ice-shelf complexes that extend into the Arctic Ocean. However, there are several physical constraint that impede the formation of a thick basin-covering Arctic Ocean ice shelf. To begin with, an ice shelf floating freely into the ocean accelerates and thins rapidly with increasing distance from the grounding line. As the



ice shelf thins it becomes susceptible to calving and tends to break up (Benn et al., 2007)[2]. Based on contemporary Antarctic analogies, glacial Arctic Ocean ice shelves are expected to have moderate extents and not reach across the entire basin (Kirchner et al., 2013). Further, for freely expanding ice shelves, the ice-mass flux across the grounding line increases rapidly with the grounding line depth (Schoof, 2007a). As a result, steady-state conditions require that the ice thickness at the grounding line

is small compared to that of the inland ice sheet; otherwise the grounding line rapidly retreats shoreward to a shallower depth where the mass flux at the grounding line matches the mass flux from the grounded ice sheet. Evidently, the observations of quasi-equilibrium glacial grounding-line depths of 1000 m in the Arctic Ocean (Jakobsson et al., 2010, 2016) are incompatible with the notion of a freely expanding ice shelf, as this would imply the existence of unrealistically large continental ice sheets.

     However, if an ice shelf is embayed or partly grounded on bathymetric highs, back stresses develop that reduce longitudinal

stresses at the grounding line and thereby buttress the grounded marine ice (Thomas, 1973; Goldberg et al., 2009). Back stresses also counteract the downstream acceleration in the ice shelf, which supports formation of thicker shelf ice. In the Arctic Ocean, embayment towards the Fram Strait is likely to provide the critical back stress that supports the growth of a basin-covering ice shelf (Grosswald and Hughes, 1999). Grounding on the Lomonosov Ridge, which is the main bathymetric high in the Central Arctic Ocean, cannot provide back stresses during the initial ice shelf growth: The Lomonosov Ridge is at its shallowest located

at around 600 m below present sea level and will therefore not cause a thinner ice shelf to ground. A fully-developed kilometre thick ice shelf, however, will ground on the Lomonosov Ridge, thereby changing dynamically in response to the developing stresses.

     Several investigators have pointed out that an ice shelf can develop from the Arctic Ocean sea-ice cover (e.g. Broecker, 1975; Hughes, 1987). A promoting factor is that Arctic Ocean wind-driven sea-ice export becomes negligible when sea-ice

cover becomes sufficiently thick and heavy (Hibler et al., 2006). During glacial conditions, grounded ice sheets will eventually develop over the Barents Sea, which further hider sea-ice export (Broecker, 1975). Accordingly, once the glacial sea ice reaches a critical thickness where the ice export becomes negligible, the sea-ice thickness will continue to grow due to surface snow accumulation and basal freezing. In addition, influx of rafted ice bergs from the continental ice sheets promotes ice growth and the emerging ice mélange increases the resistive stresses in the floating ice (Amundson et al., 2010). This processes gradually

transform the sea ice to shelf ice. The ice shelves north of Ellesmere Island today constitute Holocene examples of how such processes have created ice shelves from what most likely first was land fast ice (Antoniades et al., 2011).

     Another potential obstacle to build a thick Arctic Ocean ice shelf is the potential inflow of relatively warm water of Atlantic origin, which today occupies the basin in the depth range from about 300 to 700 m (Rudels, 2012). Studies of Antarctic ice shelves show that basal melting increases with ocean temperature and can exceed tens of meters per year (Jacobs et al.,

2011; Rignot et al., 2013). Generally, oceanic basal melting is a leading order component of the mass balance for Antarctic ice shelves (Alley et al., 2005; Pollard and DeConto, 2009), which can limit the extent or even suppress the formation of ice shelves. Further studies are needed to decide whether a thick Arctic Ocean ice shelf can be formed without radically altering the present-day features of the Arctic Atlantic Water (Lindstrom and MacAyeal, 1986).

---

[2]Numerical experiments due to Colleoni et al. (2016a) suggest that ice shelves growing from the continents can eventually cover the entire Arctic Ocean if calving is suppressed.





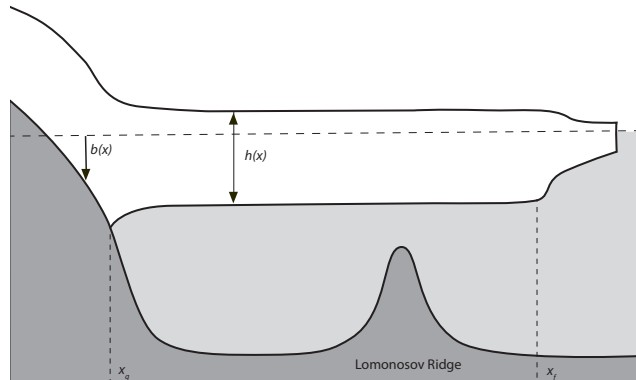

**Figure 3.** The geometry of the one-dimensional Arctic ice-sheet–ice-shelf model, illustrated in a case without grounding on the Lomonosov Ridge. Here, $h(x,t)$ denotes the ice thickness and $b(x)$ the bed elevation, counted positive below the sea level. The ice shelf has it continental grounding line at $x_g$ and is dammed towards the Fram Strait entrance, located at $x_f$, which divides the interior and exit regions of the ice shelf. The $x$-$z$ transect shown here corresponds to central $x$ transect indicated in Fig. 2. In the interior ice-shelf region $h$ is essentially constant. Near the continental grounding line ($x_g$), however, there is a narrow transition zone where the ice thickness increases away from the grounding line; see Eq. (13) and Fig. 4. In the

## 3 Analyses of ice-sheet–ice shelf dynamics based on a one-dimensional model

We will here use a one-dimensional ice dynamical model to theoretically examine the ice flow in a basin-covering Arctic Ocean ice shelf. In particular, we will consider the dynamics controlling the ice-thickness distribution and how it is affected by embayment towards the Fram Strait and by grounding on the Lomonosov Ridge. In the model, the grounded ice is assumed to be in a regime of rapid sliding in which the vertical velocity shear is small and neglected. Figure 3 illustrates the model geometry: The $x$-coordinate is aligned in the flow direction and the ice thickness along the flow is denoted $h(x,t)$.

Our analyses will show that the Arctic Ocean ice shelf has two dynamically distinct regions: one interior region, representing the main part of the Arctic Basin, and one exit region extending from the northern boarder of the Fram Strait towards a calving front in the Nordic Seas (Fig. 2). The focus of our analysis is on the interior region, where lateral stresses are assumed to be small and the ice thickness fairly uniform. The dynamics of the exit region will be represented as boundary condition for the interior region. The strongly embayed interior region is in a dynamical sense simpler, but rather different from most Antarctic ice shelves. The exit region that terminates at a broad calving front, on the other hand, share dynamical characteristics with the large Antarctic ice shelves.

### 3.1 Governing equations

To set the stage for the forthcoming analysis, the governing equations describing ice-sheet–ice-shelf dynamics are summarised here. They have been derived and discussed in detail in the literature (Muszynski and Birchfield, 1987; Schoof and Hewitt, 2013), but it is worth to emphasise a few aspects of particular importance for the case considered here. To begin with, because





the ice-shelf complex is taken to be one-dimensional in the along-flow direction and lateral stresses are ignored, the momentum balance for the floating part can be integrated horizontally. The resulting integration in combination with an upstream boundary condition, specify a relation between local ice-shelf thickness and ice-velocity divergence [Eq. (8)]. The integral constraint, set by the upstream boundary condition, depends on whether the ice shelf expands freely into the ocean or if it is embayed or re-grounds, in which case back stresses develop in the ice shelf.

Let us now proceed to present the equations. The continuity equation of the ice-sheet–ice-shelf complex is given by

$$\frac{\partial h}{\partial t} + \frac{\partial (uh)}{\partial x} = a, \tag{1}$$

where $u$, taken to be positive, is the ice velocity, $h$ the ice thickness, and $a(x,t)$ the net accumulation rate including basal freeze or melt. The grounded ice sheet obeys the vertically-integrated momentum balance

$$2\frac{\partial}{\partial x}\left(hA^{-1/n}\left|\frac{\partial u}{\partial x}\right|^{1/n-1}\frac{\partial u}{\partial x}\right) - Cu^m - g\rho_i h\frac{\partial (h-b)}{\partial x} = 0. \tag{2}$$

Here $g$ is the acceleration of gravity, $\rho_i$ the density of ice, $A$ a vertically averaged coefficient in Glen's flow law and $n \approx 3$ an associated exponent (Schoof and Hewitt, 2013). Further, $C$ and $m$ are positive constants used to model the basal frictional stress $Cu^m$.

The ice begins to float at the continental grounding line ($x = x_g$), where the flotation criteria requires that

$$h(x_g, t) = b(x_g)\rho_0/\rho_i, \tag{3}$$

where $\rho_0$ is the density of sea water. Two additional grounding lines may exist on the Lomonosov Ridge, as will be described below. There is no basal stress acting on the ice shelf, which implies that the counterpart of the momentum balance Eq. (2) for the floating ice shelf becomes

$$2\frac{\partial}{\partial x}\left(hA^{-1/n}\left|\frac{\partial u}{\partial x}\right|^{1/n-1}\frac{\partial u}{\partial x}\right) - g\rho_i\frac{\Delta\rho}{\rho_0}h\frac{\partial h}{\partial x} = 0, \tag{4}$$

where $\Delta\rho \equiv \rho_0 - \rho_i$ is the density difference between sea water and ice. Equation (4) can be integrated in $x$: We integrate from an upstream location ($x = x_u$) and obtain

$$2A^{-1/n}\left(h\left|\frac{\partial u}{\partial x}\right|^{1/n-1}\frac{\partial u}{\partial x}\right) = g\rho_i\frac{\Delta\rho}{2\rho_0}(h^2 - h_b^2). \tag{5}$$

Here, we have introduced the integration constant

$$h_b^2 \equiv \left[h^2 - h\mu^{-1}\left|\frac{\partial u}{\partial x}\right|^{1/n-1}\frac{\partial u}{\partial x}\right]_{x=x_u}, \tag{6}$$

and a parameter that depends on material properties

$$\mu \equiv g\rho_i(1 - \rho_i/\rho_0)A^{1/n}/4. \tag{7}$$





Equation (5) describes the vertically-integrated balance between the longitudinal deviatoric stress (the left-hand side) and the hydrostatic pressure modified by the upstream boundary condition related to $h_b^2$ (the right-hand side). The ice-shelf dynamics can be described succinctly by re-arranging Eqs. (5,6) to yield a relation between ice-velocity divergence and ice thickness:

$$\frac{\partial u}{\partial x} = \mu^n \left( h - \frac{h_b^2}{h} \right) \left| h - \frac{h_b^2}{h} \right|^{(n-1)}.$$ (8)

This formula can be viewed as a one-dimensional limit of a more general strain-rate relation for two-dimensional ice-shelf flow derived by Thomas (1973). By drawing on this analogy, the integration constant $h_b$, which has the dimension of length, can be connected to the ice-shelf back stress $\sigma_b$ (Thomas, 1973) as

$$\sigma_b = g\rho_i \frac{\Delta\rho}{2\rho_0} \frac{h_b^2}{h}.$$ (9)

Ice-shelf back stress (or back pressure) arises if an ice shelf runs aground on bathymetric pinning points or experiences stresses from embaying side boundaries (Thomas, 1973; Thomas and MacAyeal, 1982; Goldberg et al., 2009; Borstad et al., 2013). We emphasise that in the one-dimensional case, the back stress depends only on the local ice-shelf thickness and the global parameter $h_b$, which is set by upstream conditions [Eq. (6)]. For two-dimensional ice-shelf flows, on the other hand, the back stress depends on local ice thickness as well as on lateral and shear strain rates (Thomas, 1973; Borstad et al., 2013). This implies that the back stress defined by Eq. (9), although analogous to the back stress in two-dimensional flows (Thomas, 1973), has some unusual features that will be discussed below.

Back stress has important consequences for the dynamics of the grounded ice as well as the floating ice. Importantly, the back stress reduces the longitudinal stress at the grounding line; an effect referred to as ice-shelf buttressing. Borstad et al. (2013) introduced a non-dimensional measure of buttressing that accounts for spatial variations in an ice shelf

$$f_2(x,y,t) \equiv \sigma_b(x,y,t) \left[ \frac{g\rho_i\Delta\rho}{2\rho_0} h(x,y,t) \right]^{-1}.$$ (10)

By using Eq. (9), we can define an analogous one-dimensional buttressing parameter:

$$f(x,t) \equiv \left[ \frac{h_b(t)}{h(x,t)} \right]^2.$$ (11)

Evaluated at the grounding line, i.e. $f(x_g,t)$, this parameter becomes similar to the buttressing parameter introduced by Dupont and Alley (2005). For a freely floating ice shelf, the back stress is identical to zero, i.e $h_b = 0$ and $f = 0$. This yields minimal buttressing of the grounded ice. The limit where the grounding-line ice thickness equals the back-stress thickness $h_b$, implying that $f(x_g,t) = 1$, can be viewed as a fully-buttressed case, where the longitudinal grounding-line stress vanishes.

In summary, the dynamics of the floating ice can be described by the continuity equation (1), the floatation criteria (3), and strain relation (8). In addition, one needs to determine the global back-stress thickness $h_b$ (6) and to specify the ice mass flux at the grounding line and the upstream integration point $x_u$. Below, we will show that the global back-stress thickness $h_b$ can be determined the interior ice-shelf region.





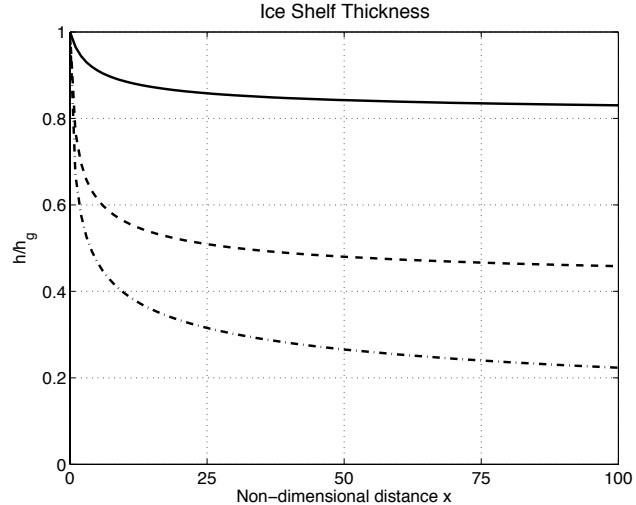

**Figure 4.** Steady state ice-shelf profiles in the limit of zero net ablation, given by Eq. (13). The depth is normalised with the grounding-line depth $h_g$ and the non-dimensional distance is defined as $x(\mu^3 h_g^3/u_g)$. The solid, dashed, and the dashed-dotted lines represent $h_b/h_g = 0.8$, $h_b/h_g = 0.4$, and $h_b/h_g = 0.0$, respectively. At large distances, the ice-shelf thickness approaches the back-pressure thickness $h_b$, implying that without back pressure ($h_b = 0$) the ice-shelf thickness shrinks to zero far away from the grounding line.

## 3.2 Qualitative analysis of ice-shelf dynamics

The governing equations are now analysed with focus on the dynamics of the ice shelf inside of the Fram Strait. First, we derive analytical steady-state ice-shelf profiles in the limit of vanishing accumulation, which approximately describe the transition zone between the grounding line and the central ice shelf. We then go on to consider time-evolving dynamics. This scenario is examined using a scaling analysis, in which the equations are put on non-dimensional form and small non-dimensional parameters are identified: the resulting equations describe the interior ice shelf to the lowest order (Fig. 2).

As the ice becomes afloat at the grounding line, the ice velocity accelerates and the ice thickness decreases. Here, the rates of ice divergence are typically large compared to the net accumulation $a$. As a results, the ice thickness approximately decreases offshore initially as if the ice mass flux would be constant. To examine the dynamics in this limit, we assume steady-state conditions and neglect accumulation, i.e. $a = 0$ implying that $q = uh$, where $q$ is a constant ice mass flux. If $n = 3$ as usually chosen in Glen's flow law, a simple analytical solution can be obtained. By using the fact that $u = q/h$, Eqs. (1,8) can be combined and after straightforward calculations be rewritten as

$$\frac{\partial h^2}{\partial x} = -\frac{2\mu^3}{q}\left(h^2 - h_b^2\right)^3. \tag{12}$$

This equation can be integrated in terms of $h^2$ and after a few manipulations one obtains

$$h(x) = h_g \left[\gamma + \left((1-\gamma)^{-2} + 4x(\mu^3 h_g^4/q)\right)^{-1/2}\right]^{1/2}, \tag{13}$$





where $h_g$ is the grounding line depth and we have defined

$$\gamma \equiv (h_b/h_g)^2. \tag{14}$$

Here, we require that $h_g > h_b$ and note that the limit $h_g = h_b$ corresponds to a fully-buttressed case. If there is no back
pressure, i.e. $h_b = 0$, the free-spreading shelf profile results (Oerlemans and van der Veen, 1984). In this case, the longitudinal

deviatoric stress exactly balances the hydrostatic pressure [see Eq. (5)], and the ice-shelf thickness shrinks towards zero at
large distances. Figure 4 shows that with back pressure, the ice-shelf thickness instead asymptotically approaches $h_b$ far away
from the grounding line. As a consequence, the longitudinal stress decreases towards zero with increasing distance from
the grounding line. The natural length scale of the problem is $u_g/(\mu h_g)^n$. Using the values $n = 3$, $A = 1 \cdot 10^{-25}$ Pa$^{-3}$ s$^{-1}$,
$h_g = 1000$ m, and $u_g = 0.1$ km year$^{-1}$, we find that $u_g/(\mu^3 h_g^3) \sim 3$ km. Figure 4 shows that with back pressure, $h \approx h_b$ at

non-dimensional distances $x(\mu^3 h_g^3/u_g)$ on the order of 30 or larger. For the present parameters, one finds that an ice shelf with
back pressure has essentially a uniform thickness $h_b$ a few 100 kilometres away from the grounding line. By examining the
non-dimensional equations presented below, it can be shown that Eq. (13) approximately describes the ice-shelf thickness in
a transition zone with a width on the order of $u_g/(\mu h_g)^n$ also when the accumulation is non zero and the grounding line and
central basin ice thicknesses evolve slowly.

The length scale of a glacial Arctic Ocean ice shelf (say $L$) is typically much larger than the intrinsic ice-shelf length scale
$u_g/(\mu h_g)^n$, and their ratio defines a small non-dimensional parameter

$$\epsilon \equiv \frac{u_g}{(\mu h_g)^n L}. \tag{15}$$

The scaling analysis of Schoof (2007b) shows that $\epsilon^{1/n}$, based on ice sheet characteristics, measures the ratio between the
longitudinal stress and the hydrostatic pressure in the grounded ice. By taking $L \sim 1000$ km, we find that $\epsilon \sim 10^{-2}$ or smaller

for a kilometre thick Arctic Ocean ice shelf. To exploit the fact that $\epsilon$ is small, we put the equation on non-dimensional form.
We denote the non-dimensional variables with an asterisk and introduce the following scales

$$u_* = u/u_g, \; h_* = h/h_g, t_* = t(u_g/L), \; x_* = x/L, \; a_* = aL/(u_g h_g). \tag{16}$$

Using these scales in Eqs. (1,8) yield the following non-dimensional equations

$$\frac{\partial h_*}{\partial t_*} + u_* \frac{\partial h_*}{\partial x_*} + h_* \frac{\partial u_*}{\partial x_*} = a_*, \tag{17}$$

$$\epsilon \frac{\partial u_*}{\partial x_*} = (h_* - \gamma/h_*) |h_* - \gamma/h_*|^{(n-1)}. \tag{18}$$

To the lowest order in $\epsilon$, Eq. (18), yields $h_* \approx \sqrt{\gamma}$; or on dimensional form

$$h(x,t) = h_b(t). \tag{19}$$





Notably, the interior ice-shelf thickness is approximately constant and equal to the back-pressure thickness $h_b$. The approximate uniformity of the ice thickness follows also from the non-dimensional ice-shelf momentum balance [Eq. (4)]

$$\frac{\epsilon^{1/n}}{2}\frac{\partial}{\partial x_*}\left(h_*\left|\frac{\partial u_*}{\partial x_*}\right|^{1/n-1}\frac{\partial u_*}{\partial x_*}\right) - h_*\frac{\partial h_*}{\partial x_*} = 0,$$ (20)

which to the lowest order in $\epsilon$ predicts a constant ice-shelf thickness. Since the ice-shelf thickness is approximately equal to $h_b$, the buttressing parameter $f(x,t)$ approaches unity in the bulk of the ice shelf. This can be contrasted with the behaviour in most Antarctic ice shelves, where the buttressing and back stress generally decrease seaward towards calving fronts (Thomas and MacAyeal, 1982; Borstad et al., 2013). The difference arises because we are here considering a strongly confined segment of the interior Arctic Ocean ice shelf, which terminate dynamically north of the Fram Strait and/or at span-wise extensive grounding line on the Lomonosov Ridge. In the exit region near the Fram Strait, the patterns of back stress and buttressing should resemble those observed in less strongly embayed Antarctic ice shelves such as the Ross and Larsen C (Thomas and MacAyeal, 1982; Borstad et al., 2013).

The leading-order physics governing the interior ice-shelf segment can be summarised as follows:

1. The dominance of the hydrostatic pressure gradient in the momentum balance forces the ice-thickness to be nearly uniform. As a result, local variations in net accumulation are forced to be balanced by variations in ice velocity divergence, which have a negligible expression in ice thickness.

2. Since the ice-shelf thickness is constant, we can integrate the mass-balance equation (1) from the grounding line to an upstream point $x_u$ to obtain

$$(x_u - x_g)\frac{\partial h_S}{\partial t} = \int_{x_g}^{x_u} a(x,t)\,dx + q(x_g,t) - q(x_u,t),$$ (21)

where $h_S \approx h_b$ is the mean Arctic Ocean ice-shelf thickness[3], $q(x_g,t)$ and $q(x_u,t)$ the ice mass fluxes at the continental grounding line and the upstream point $x_u$, respectively. Accordingly, the interior ice-shelf thickness is to the lowest controlled by the horizontally-integrated net accumulation and is hence fairly insensitive to the spatial distribution of the accumulation.

In section 4, we will consider how suitable upstream integration points are selected depending on ice-shelf geometry. However, it is worth to note a principle difference between the cases where the ice shelf grounds on the Lomonosov Ridge and where back stresses arise from embayment towards the Fram Strait. In the former case, the integration point is well defined by the grounding line position on the ridge. In the latter case, there is no well defined upstream integration point, but rather a transition zone near the Fram Strait where ice-shelf flow shifts between two dynamical regimes (Fig. 2). We will take a pragmatic view and assume that an upstream integration point can be selected slightly upstream of the transition zone, where

---

[3]For notational clarity we use a basin-mean ice thickness $h_S$, rather than $h_b$, in the mass balance: when the shelf grounds on the Lomonosov Ridge there will be different back pressures in the Amerasian and Eurasian parts of the shelf; see below.





the viscous stresses are negligible to the lowest order. Here, the ice-thickness gradients are small implying that the resulting value of the back-stress thickness $h_b$ is insensitive to the exact location of the integration point.

We note that also for two-dimensional ice-shelf flows, an analogous scaling analysis indicates that the ice-shelf thickness should be approximately uniform when the parameter $\epsilon$ is small (Goldberg et al., 2009). Thus, provided that the geometry supports strong embayment, we anticipate that the present one-dimensional scaling results hold qualitatively for more realistic ice-shelf configurations. Indeed, two-dimensional numerical simulations of strongly embayed, thick ice shelves with large horizontal extents (Lindstrom and MacAyeal, 1986; Colleoni et al., 2016a) support the notion that the thickness of an Arctic Ocean ice shelf should, to the lowest order, be horizontally uniform in the central basin.

### 3.3 The grounding line and Fram Strait mass fluxes

Next, we turn to two key ice mass fluxes that control the mass balance of the interior Arctic Ocean ice sheet-shelf complex: the mass flux at the continental grounding line and the mass flux through the Fram Strait. The grounding-line mass flux presented here can be derived using a boundary layer approach (Schoof, 2007b). In contrast, the ice mass flux through the Fram Strait has a more complex dependence on local bathymetry and is less amendable to theoretical analyses. We will use an analytic model developed by Hindmarsh (2012) to discuss the Fram Strait dynamics in qualitative terms.

Schoof (2007b) derived a formula for the grounding-line mass flux in the limit where the longitudinal stress is small compared to the hydrostatic pressure in the ice sheet, i.e. the parameter $\epsilon$ [Eq. (15)] based on the ice-sheet characteristics is taken to be small. As shown by Schoof (2007a), the mass-flux formula can be modified to account for a reduction in the grounding-line stress due to back pressure[4]:

$$q_g(h_g, h_b) = \left[ (\rho_i g \mu^n / C)^{\frac{1}{m+1}} h_g^{(m+n+3)/(m+1)} \right] \left( 1 - \frac{h_b^2}{h_g^2} \right)^{\frac{n}{m+1}}, \qquad (22)$$

where $h_g$ is the local grounding-line ice thickness and $h_b$ the back pressure thickness. The terms within the square brackets are the asymptotic mass-flux formula derived by Schoof (2007b) for a freely spreading ice shelf. The last factor accounts for the grounding-line stress reduction due the back stress. This term is a function of $1 - f(x_g, t)$ and hence reflects buttressing; see Eq. (11). Further, a positive ice flux requires that $h_g > h_b$. The key qualitative feature of Eq. (22) is that the mass flux at the grounding line, to the lowest order, depends only on the local ice thickness $h_g$ and the back pressure thickness $h_b$. Without back pressure, the mass flux increases rapidly with the grounding line depth; for the standard values $n = 3$ and $m = 1/3$ on finds that $q_g \sim h_g^5$. This forces the grounding-line thickness of a freely spreading shelf to be small compared to the thickness of the inland ice sheet (Schoof, 2007b).

In the ice-shelf exit regions towards and in the narrow Fram Strait, mass conservation requires increasing ice velocities and there will also be stronger lateral stresses from grounded ice along the strait margins (Goldberg et al., 2009; Hindmarsh, 2012). Stress divergence in the strait is balanced by a hydrostatic pressure gradient associated with a downstream thinning of the ice shelf towards the calving front. The ice-shelf flow will be two-dimensional and interact with the rather complex

---

[4]This corresponds to Eq. (29) in Schoof (2007a), and results if our expression for $\partial u / \partial x$ [Eq. (8)] is used in the derivation given in section 3.1 of that paper.





bathymetry in the Fram Strait area (Fig. 1a). However, it is conceivable that there exists an approximate Fram Strait ice-volume export relation, which depends mainly on the strait geometry and the mean ice-shelf thickness in Arctic Ocean. To examine an idealised flow regime, we follow ideas presented by Hindmarsh (2012) and view the Fram Strait as a straight channel with a constant width $L_y$. Sufficiently far away from the calving front, a Poiseuille-like ice flow assumed, for which there is a balance

between the lateral stress divergence and the hydrostatic pressure gradient

$$2\frac{\partial}{\partial y}\left(h(2A)^{-1/n}\left|\frac{\partial u}{\partial y}\right|^{1/n-1}\frac{\partial u}{\partial y}\right) - g\rho_i(1-\rho_i/\rho_0)h\frac{\partial h}{\partial x} = 0, \tag{23}$$

where $y$ is the cross-channel coordinate. If the ice shelf thickness is constant in the cross-channel direction and the ice velocity is zero at the channel boundaries, integration of Eq. (23) yields the following along-stream velocity

$$u(x,y,t) = L_y\left(-2L_y\mu\frac{\partial h}{\partial x}\right)^n \frac{\left(1-|y/L_y|^{n+1}\right)}{(n+1)}, \tag{24}$$

where $\mu$ is defined in Eq. (7). By assuming that $uh$ is constant along the stream lines in the strait and that $\partial h/\partial x \sim h_S/L_x$, where $h_S$ is the ice thickness upstream in the Arctic Ocean and $L_x$ is the along-stream strait extent, the ice volume flux $Q_f$ scales as

$$Q_f(h_S) = \int_{-L}^{L} uh\,dy \propto L_y^2\left(\frac{L_y\mu}{L_x}\right)^n h_S^{n+1}. \tag{25}$$

In the one dimensional frame work, one uses $q_f(h) = Q_f(h)/L_{up}$, where $L_{up}$ is the width of the upstream Arctic Basin. The

qualitative mass-flux relation given in Eq. (25) has the same dependence on the ice thickness as the mass flux in the analytical model of Hindmarsh (2012); see section 5.3 in his paper. For $n = 3$, one finds that $q_f(h_S) \propto h_S^4$. This suggests that the Fram Strait ice export increases with upstream ice thickness at a rate that is faster than linear. However, interactions between the ice shelf and rather the complex strait bathymetry may yield an ice export relation that differs qualitatively from Eq. (25). For instance, when the ice shelf becomes thick enough to ground on the broad Yermak Plateau just north of the Fram Strait (Fig.

1a), basal stresses will increase abruptly. To incorporate such effects into Eq. (25) would require that the effective Fram Strait width $L_y$ and length $L_x$ become functions of the ice-shelf thickness. We do not consider this problem here, but emphasise the uncertainties associated with Eq. (25).

### 3.4 The ice mass flux over the Lomonosov Ridge

Consider next what happens when the ice shelf re-grounds and un-grounds over the Lomonosov Ridge, as outlined in Fig. 5.

The multi-beam data reported by Jakobsson et al. (2016) reveals ice-erosion patterns on the higher parts of the Lomonosov Ridge created by a coherent flow of grounded ice from the Amerasian to the Eurasian Basin. The stresses from the grounding on the Ridge would have increased gradually from the point when the ice shelf initially intercepted the highest part of the Ridge to the latter stages when the geophysical data suggests that the ice shelf on the upstream side was grounded more than 1200 m below the present day sea level. We will here analyse a one-dimensional scenario in which the Lomonosov Ridge is viewed as





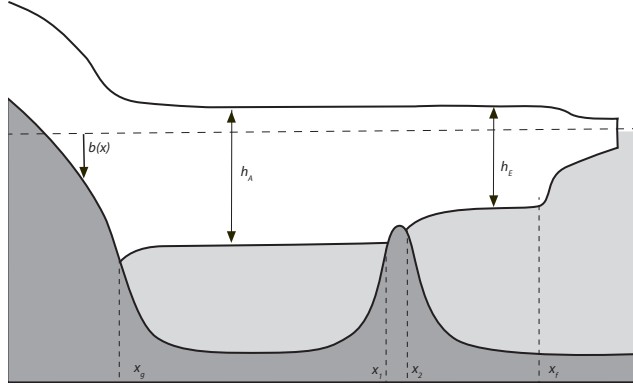

**Figure 5.** The Arctic Ocean ice-shelf geometry with grounding on the Lomonosov Ridge. The mean ice thicknesses in the Amerasian Basin (to the left of the Ridge) and the Eurasian Basin (to the right of the Ridge) are denoted $h_A$ and $h_E$, respectively. The grounded-ice extent on the ridge is assumed to be long enough for an ice-sheet like flow to develop. The upstream and downstream grounding-line positions on the Ridge are denoted $x_1$ and $x_2$, respectively. At $x_1$, the ice-shelf thickness is $h_A$ and for notational convenience we introduce $h_2 \equiv h(x_2, t)$.

bathymetric pinning point that extends over the entire ice-shelf width. The dynamics in this idealised scenario is different from the case where local pinning point intercept only a fraction of the lateral ice-shelf extent: Ice tends to flow around rather than over local pinning points, which yield two-dimensional ice flows associated with significant lateral stresses (Goldberg et al., 2009; Favier et al., 2012; Borstad et al., 2013). We underline that the Lomonosov Ridge blocked most but not all of the ice-shelf

stream (Fig. 1), implying a partly two-dimensional flow with some of the cross-ridge ice flux occurring over deeper channels with ungrounded ice. Thus, the following one-dimensional analysis provides only a qualitative description of the interactions between the ice shelf and the Lomonosov Ridge.

    Our goal here is to obtain a relation between the ice mass flux over the Lomonosov Ridge and the ice thicknesses at the upstream and downstream grounding lines (Fig. 5). We anticipate that the basal stress in the grounded ice is primarily balanced

by a hydrostatic pressure gradient associated with a downstream slope of the ice surface. In transition zones near the two grounding lines, however, the longitudinal deviatoric stress is generally important in the momentum balance and local ice rumples may form (Schoof, 2007b, 2011; Goldberg et al., 2009; Favier et al., 2012). The non-dimensional width of these boundary layers are controlled by the parameter $\epsilon$ based on the ridge-flow characteristics (Schoof, 2007b): when this parameter is small the horizontal extent of the boundary layers are small compared to that of the grounded-ice segment on the Lomonosov Ridge.

The multi-beam data shows stream-wise grounding distances on the order of 20–40 km (Jakobsson et al., 2016), indicating that $\epsilon \sim 0.1$; see section 3.2. Although the value of $\epsilon$ is only moderately small, we will here neglect the dynamics in the transitional boundary layers and assume that the leading-order momentum balance on the Lomonosov Ridge is given by

$$-Cu^m - \rho_i gh\frac{\partial s}{\partial x} = 0, \ s \equiv h - b. \tag{26}$$

where $s$ is the ice surface $s$. As illustrated in Fig. 6, the down-stream decrease in $s$ required to balance the basal stress causes

the upstream grounding line on the Ridge to lay deeper that the downstream one.





To obtain a rough estimate on the difference in grounding-line depths across the ridge, we re-write Equation (26) as

$$\frac{\partial s}{\partial x} = -\frac{Cq^m}{\rho_i g h^{m+1}},\tag{27}$$

where $q = uh$. The flotation criteria implies that

$$s(x_1, t) - s(x_2, t) = (h_1 - h_2)\Delta\rho/\rho_0, \; h_{1/2} \equiv h(x_{1/2}, t)\tag{28}$$

where $h_1$ and $h_2$ denote the grounding-line ice thicknesses and $\Delta\rho \equiv \rho_0 - \rho_i$. Treating $h$ as a constant in Eq. (26), we obtain the following estimate

$$\frac{h_1 - h_2}{h_r} \sim \phi, \; \phi \equiv \frac{\rho_0 C L_r q^m}{g \Delta\rho \rho_i h_r^{m+2}},\tag{29}$$

where $h_r$ is the ice thickness at the ridge crest and $L_r$ the length of grounded ice on the ridge. The non-dimensional parameter $\phi$ can be viewed as a measure of the flow resistance of the ridge: taking $h_r = 1$ km, $L_r = 20$ km, $u = 0.1$ km year$^{-1}$, $m = 1/3$,

and $C = 7.6 \cdot 10^6$ Pa m$^{-1/3}$ s$^{1/3}$, one obtains $\phi \sim 1$. The multi-beam estimated difference in grounding-line depths across the Ridge, which is a few hundreds of meters, suggests that $(h_1 - h_2)/h_r \sim 0.3$. Given the approximate nature of the scaling, these two estimates should be viewed as similar and thus mutually compatible.

At the upstream grounding line in the Amerasian Basin, there is no transition layer in the ice shelf as such a layer only develops downstream of a grounding line; see section 3.2. Accordingly, the ice thickness there essentially equals the mean

ice-shelf thickness in the Amerasian Basin $h_A$. The flotation criteria Eq. (3) specifies the upstream grounding-line position $x_1$, yielding a relation of the form

$$x_1 = x_1(h_A, b(x)).\tag{30}$$

Near this grounding line, there will be a boundary layer in the grounded ice that matches smaller values of $\partial u/\partial x$ in the floating ice with higher ones closer to the ridge crest, where the ice thins and the velocity increases.

A simplified description of the ice thickness over the Lomonosov Ridge is obtained by assuming that the variations in $h$ are dominated by the variations of the sea bed $b(x)$ and that $q$ is constant. Using this and integrating Eq. (27) from $x = x_1$, one obtains

$$h(x, t) = h_A \Delta\rho/\rho_i + b(x) - \frac{Cq^m}{\rho_i g} \int_{x_1}^{x} \frac{dx'}{b(x')^{m+1}}\tag{31}$$

where we have used the flotation criteria $h_A(t) - b(x_1) = h_A(t)\Delta\rho/\rho_i$. From Eq. (31), the flotation criteria specifies the

downstream grounding line depth and position

$$h(x_2, t) = \rho_0/\rho_0 b(x_2), \; x_2 = x_2(h_A, q, b(x)).\tag{32}$$

Figure 6 illustrates the ice thickness over the Ridge obtained from Eq. (31) for a parabolic ridge:

$$b(x) = b_r(1 + (x/L_r)^2),\tag{33}$$



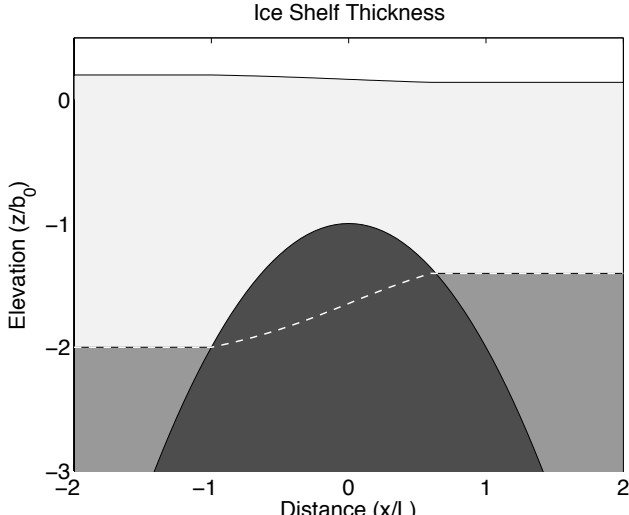

**Figure 6.** A sketch of the ice-thickness distribution near the Lomonosov Ridge, where the ice flows from the left (Amerasian side) to the right (Eurasian side). A parabolic ridge shape is used [Eq. (33)] and the ice thickness is given by Eq. (31), which assumes that the longitudinal deviatoric stress is negligible in the grounded ice. The vertical coordinate is normalised with $b_r$ the ridge-crest depth. The white dashed line indicates the base of floating ice, which mirrors the ice-surface elevation. Near the two grounding lines, there will be narrow transitions zones where the longitudinal deviatoric stress is not negligible and local ice rumples can form (Schoof, 2007b; Goldberg et al., 2009; Favier et al., 2012). However, the depicted simple solution should give the approximate difference in grounding-line depths across the ridge. Note that a transition layer downstream of the the ridge where the ice shelf thins is not shown here; see Fig. 4.

where $b_r$ is the ridge-crest depth and $L_r$ a horizontal length scale.

The relations (31) and (32) can be used to calculate a Ridge mass-flux relation on the form

$$q_r = q_r(h_A, h_2), \tag{34}$$

which implicitly depends on bed shape $b(x)$. For a parabolic ridge [Eq. (33)], an approximation of $q_r$ is obtained by taking

5  $b(x) \approx b_r$ in the integral in Eq. (31) and then using Eq. (32). After some algebra, one obtains the downstream $x_2$ grounding-line thickness

$$h_2(h_A, q) = h_A + h_r[\phi^2 - 2\phi(h_A/h_r - 1)^{1/2}], \quad h_r \equiv b_r\rho_0/\rho_i, \tag{35}$$

where $\phi \sim q^m$ is defined in Eq. (29). The upstream $x_1$ grounding-line thickness is given by

$$h_A(h_2, q) = h_2 + h_r[\pm 2\phi(h_2/h_r - 1)^{1/2} + \phi^2], \tag{36}$$

10  where the plus/minus sign specifies the solution with the $x_2$ grounding line downstream/upstream of the ridge crest. Solving for $\phi$, we obtain

$$\phi(h_A, h_2) = (h_A/h_r - 1)^{1/2} \pm (h_2/h_r - 1)^{1/2}, \tag{37}$$





where the plus/minus sign now specifies the solution with the $x_2$ grounding line upstream/downstream of the ridge crest. This relation and Eq. (29) specifies the ridge mass flux: $q_r \sim \phi^{1/m}$.

In the example above, the grounded-ice extent and hence the effective basal stress vary with the strength of the mass flux. This makes the relation between mass flux and grounding-line thicknesses more complicated than in the simple scaling given in Eq. (29) where the the grounded-ice extent is taken to be fixed; in this case one obtains $q_r \sim (h_A - h_2)^{1/m}$. For a simple symmetric ridge geometry, however, the qualitative features of the mass-flux relation Eq. (34) are straightforward to describe; see Fig. (7). In the limit where the mass flux $q$ is vanishingly small, the ice is symmetric with respect to the ridge crest and $h_2 = h_A$. As $q$ increases, the drop in surface ice elevation over the Ridge increases; see Eq. (27). If the upstream grounding-line thickness $h_A$ is fixed, then the downstream grounding-line thickness will decrease with increasing $q$, which implies that this grounding line will shift towards the ridge crest. Accordingly, for a given upstream ice thickness, there is a threshold value of $q$ that places the downstream grounding line on the ridge crest. For higher values of $q$, both grounding lines are located upstream of the ridge crest. In this regime, the upstream grounding-line thickness increases with increasing $q$, until the two grounding lines meet[5]. Furthermore, the sea bed at the $x_2$ grounding line now shoals in the ice-flow direction, which based on the classical analysis of marine ice sheet stability (e.g. Weertman, 1974; Schoof, 2012) indicates that this configuration may represent an unstable steady state. We will analyse this matter further in section 4.

We emphasise that the grounded ice on the Lomonosov Ridge is assumed to in a regime of rapid sliding and governed by the momentum-balance relation Eq. (2). Accordingly, the formula of Schoof (2007a) provides the mass flux on the downstream side of the Lomonosov Ridge, where the ice becomes afloat in the Eurasian Basin. Using Eq. (22), one obtains $q_g(x_2) = q_g(h_2, h_E)$, where the back pressure $h_b$ is now set by the Eurasian Basin ice-shelf thickness $h_E$. Further, as we here are considering a quasi-equilibrium flow associated with a spatially constant mass flux over the the ridge, we require the mass flux relations (22) and (34) to match

$$q_r(h_A, h_2) = q_g(h_2, h_E). \qquad (38)$$

This yields a functional relation of the form $h_2 = h_2(h_A, h_E)$, relating the downstream grounding-line ice thickness $h_2(x_2, t)$ to the mean ice thicknesses in the Amerasian and Eurasian Basins. Accordingly, we can write the ridge mass flux on the from

$$q_r = q_r(h_A, h_E), \qquad (39)$$

which depends implicitly on the functional relation $h_2 = h_2(h_A, h_E)$. A scaling analysis of Eq. (38), however, indicates that to the lowest order of approximation $h_2 \approx h_E$, which results from the high sensitivity of $q_g(h_2, h_E)$ on $h_2$. Accordingly, the quasi-equilibrium mass flux over the Lomonosov Ridge can be approximated by taking $h_2 = h_E$ in Eq. (37), which provides the mass flux relation $q_r = q_r(h_A, h_E)$. This simplification will be used in the analyses below.

---

[5]Note that when grounded-ice extent becomes smaller, the importance of the longitudinal viscous stress grows. Hence, the approximate balance (26) breaks down before the grounding lines meet, which will modify the mass-flux features in this limit.





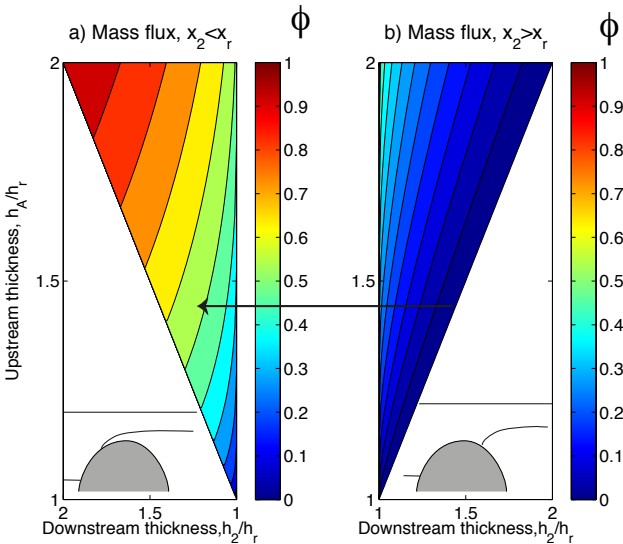

**Figure 7.** The mass-flux relation over the Lomonosov Ridge $q_r$ [Eq. (37)] as a function of the grounding-line thicknesses (upstream $h_A$ and downstream $h_2$). The mass flux, which is normalised to be unity at its maximum, is shown in terms of $\phi$; see Eq. (29). No steady states exits when $h_2 > h_A$. Panel a) shows ice-shelf configurations where both grounding lines are located on the upstream Amerasian side of the ridge crest (at $x_r$, see the sketch at bottom), where the sea bed slopes upward in the ice-flow direction. These configurations may be unstable and/or attainable in a steady state (see the text for details). Panel b) shows the configurations with grounding lines on each sides of the crest, which represent stable steady states. Note that $x$-axis in panel a) is reversed: moving along the arrow at a fixed $h_A$ one sees how the mass flux increases as the upstream $h_2$ grounding line shifts upward to the ridge crest and then downwards on the Amerasian side.

## 4 Dynamical stages of ice-shelf evolution

Based on the results of section 3, we now go on to qualitatively describe important stages in the ice-shelf evolution and considering their stability. In section 4.1, we consider a thinner ice shelf, which does not ground on the Lomonosov Ridge, and analyse its stability and coupling to surrounding ice sheets. Section 4.2 deals with the dynamics related to ice-shelf grounding

5   on the Lomonosov Ridge and analyses the stability of equilibrium states of the Arctic Ocean ice shelf complex. Here, we find that the ice sheet-shelf complex should be stable if the downstream grounding line on the Lomonosov Ridge is located on a downward sloping bed. While this is reminiscent of classical results on marine ice sheet stability (Weertman, 1974), we emphasise that the present result is derived under different conditions: in the presence of two grounding lines, of which the downstream one features a mass flux that decreases with local ice thickness; rather than increases as in the classical case.

10 **4.1 The ungrounded shelf regime**

A thinner Arctic Ocean ice shelf, which does not ground on the Lomonosov Ridge, will essentially behave as single dynamical entity characterised by a nearly uniform thickness inside of the Fram Strait. In this regime, the integrated ice-shelf mass balance





[see Eq. (21)] is given by

$$(x_f - x_g)\frac{\partial h_S}{\partial t} = \int_{x_g}^{x_f} a(x,t)\,dx + q_g(h_g, h_S) - q_f(h_S), \tag{40}$$

where $h_S(t)$ is the basin-mean ice-shelf thickness, $(x_f - x_g)$ the length of the shelf, $q_g$ and $q_f$ the mass fluxes at the grounding line and the Fram Strait entrance, respectively. The mass balance of the continental ice sheet is

$$\int_{x_0}^{x_g} \frac{\partial h(x,t)}{\partial t}\,dx = \int_{x_0}^{x_g} a(x,t)\,dx - q_g(h_g, h_S), \tag{41}$$

where $x_0$ is the position of the continental ice divide where $u = 0$. The mass balance equations describe a marine ice sheet with a moving grounding line (Schoof, 2007a), which via the grounding-line mass flux $q_g(h_g, h_S)$ is coupled to an embayed ice shelf. By assuming a steady state and adding the two mass-balance relations, we obtain

$$q_f(h_S) = \int_{x_0}^{x_f} a(x)\,dx. \tag{42}$$

This shows that in a steady state, the ice-shelf thickness is controlled by the Fram Strait ice export $q_f(h_S)$ and the net accumulation over the ice-sheet–shelf complex.

To consider some qualitative stability aspects of the ice-sheet–ice-shelf complex, we consider the two limiting cases of a large ice sheet and a large ice shelf, respectively. To begin with, assume that the ice sheet is dynamically stable and that its volume is much smaller than that of the ice shelf. In this case, the time scale of the ice shelf is much longer than that of the ice sheet and the time tendency can be ignored in Eq. (41); implying that the net accumulation over the ice sheet approximately balances the grounding-line mass flux. Adding the mass balance equations in this limit gives

$$(x_f - x_g)\frac{\partial h_S}{\partial t} = \int_{x_0}^{x_f} a(x)\,dx - q_f(h_S). \tag{43}$$

In this limit, steady states of the ice complex are stable if $\partial q_f / \partial h_S > 0$, which ensures that perturbations in ice-shelf thickness are associated with countering anomalies in the Fram Strait ice export.

The opposite limit, where time scale of the ice shelf is much shorter than that of the ice sheet, the time tendency can be ignored in the ice-shelf mass balance Eq. (40); implying an approximate balance between the net accumulation over the ice shelf, including the grounding-line mass flux and the Fram Strait ice export. The stability of steady-state solutions can be analysed using a linear stability analysis (Schoof, 2012): We consider small perturbations of the grounding-line position and the ice thicknesses from an equilibrium state:

$$x_g(t) = \overline{x}_g + x'_g(t), \; h_g(t) = \overline{h}_g + h'_g(t), \; h_S(t) = \overline{h}_S + h'_S(t), \tag{44}$$

where the overbar and the prime denote steady states and perturbations, respectively. Using this ansatz in Eqs. (40,41) and retaining only the terms that are linear in the perturbation variables, we obtain for the ice shelf

$$0 = -x'_g a(\overline{x}_g) + h'_g\left(\frac{\partial q_g}{\partial h_g}\right) + h'_S\left(\frac{\partial q_g}{\partial h_S} - \frac{\partial q_f}{\partial h_S}\right), \tag{45}$$





and for the ice sheet

$$\int_{x_0}^{\overline{x}_g} \frac{\partial h'(x,t)}{\partial t}\, dx = x'_g a(\overline{x}_g) - h'_g \left(\frac{\partial q_g}{\partial h_g}\right) - h'_S \left(\frac{\partial q_g}{\partial h_S}\right). \tag{46}$$

Here, the partial derivates of the mass fluxes are evaluated for the steady-state conditions and for simplicity the net accumulation is assumed to be continuous across the grounding line: $a(\overline{x}_g)$ is taken to be the same in the two equations above; an assumption

which is easy to relax. Equation (45) describes a quasi-equilibrium evolution of the ice shelf, in which its thickness adjust instantaneously to changes in the grounding-line position and thickness. This relates $h'_S$ to $h'_g$ and $x'_g$, allowing Eq. (46) to be re-formulated as

$$\int_{x_0}^{\overline{x}_g} \frac{\partial h'(x,t)}{\partial t}\, dx = x'_g (1-\chi) a(\overline{x}_g) - h'_g (1-\chi) \left(\frac{\partial q_g}{\partial h_g}\right), \tag{47}$$

where we have introduced

$$\chi \equiv -\frac{\partial q_g}{\partial h_S} \left(\frac{\partial q_f}{\partial h_S} - \frac{\partial q_g}{\partial h_S}\right)^{-1}. \tag{48}$$

An inspection of the Eqs. (22,25) shows that $\partial q_g/\partial h_S < 0$ and $\partial q_f/\partial h_S > 0$, which implies that $0 < \chi < 1$. In analogy with the buttressing parameter $f$ [Eq. (11)], $\chi$ can be viewed to measure the buttressing associated with time-evolving perturbations: $\chi = 0$ corresponds to minimal buttressing. This limit occurs if the Fram Strait mass flux is highly sensitive to ice-thickness variations (i.e. $\partial q_f/\partial h_S \gg -\partial q_g/\partial h_S$). In this limit, the Fram Strait constriction provides no resistance for the ice flow and

hence no buttressing of the ice sheet.

If the sea bed has a nonzero slope near the grounding line, the flotation criteria Eq. (3) specifies a unique relation between the grounding-line position and ice thickness, which yields the following linearised relations

$$h'_g \left(\frac{\partial q_g}{\partial h_g}\right)_{\overline{h}_g} = x'_g \frac{\rho_0}{\rho_i} \left(\frac{\partial b(x)}{\partial x}\right)_{\overline{x}_g} \left(\frac{\partial q_g}{\partial h_g}\right)_{\overline{h}_g} = x'_g \left(\frac{\partial q_g}{\partial x_g}\right)_{\overline{x}_g}. \tag{49}$$

By using this to rewrite the last term on the right-hand side of Eq. (47), we obtain

$$\int_{x_0}^{\overline{x}_g} \frac{\partial h'(x,t)}{\partial t}\, dx = x'_g (1-\chi) \left[a(\overline{x}_g) - \left(\frac{\partial q_g}{\partial x_g}\right)\right]. \tag{50}$$

It is the sign of the term within the square brackets that decides whether the ice sheet gains or loses mass if the grounding line advances: The analysis of Weertman (1974) suggests that the ice sheet is stable if

$$a(\overline{x}_g) - \left(\frac{\partial q_g}{\partial x}\right)_{\overline{x}_g} < 0, \tag{51}$$

which implies that advances/retreats of the grounding lines cause the ice sheet to lose/gain mass. Schoof (2012) showed that

this linear stability criteria can be derived strictly in a more complete model, which considers the dynamics of perturbations in





thickness and flow in the grounded ice sheet and how they interact with shifts of the grounding-line position. In the absence of net accumulation or ablation [i.e. $a(\overline{x}_g) = 0$], it follows directly from the Eqs. (49,51) that if the grounding-line mass flux increases with ice thickness (i.e. $\partial q_g / \partial h_g > 0$), then the ice-sheet–ice-shelf complex is stable provided that the grounding line is on a downward sloping bed (i.e. $\partial b / \partial x > 0$).

We note that since the factor $1 - \chi$ in Eq. (50) is always greater than zero, the present analysis seems to suggest that ice-shelf buttressing cannot remove the instability of a marine ice sheet on an upward sloping sea bed, but only reduce the degree of instability. This somewhat counterintuitive result stems from the fact we are considering an extensive and effectively dammed ice-shelf segment, for which influences of lateral stresses bed are ignored. In this limiting case, the back stress is set by the mean ice shelf thickness and is hence taken to be independent of distance from the continental grounding line to Fram Strait

exit region. However, back stresses generally increase when the grounding line retreats and the embayed ice shelf experiences lateral stresses over an extending distance. This buttressing effect, which tends to decrease the mass flux at the grounding-line when it retreats, can potentially stabilise a marine ice sheet with a grounding line on an upward sloping bed. Numerical simulations due to Goldberg et al. (2009), however, suggest that this mechanism only can stabilise a marine ice sheet on un upward sloping bed if the ice-shelf is laterally confined in a rather narrow embayment. Thus, we anticipate that the stability

criteria due to Weertman (1974) [Eq. (51)] applies qualitatively for Arctic marine ice sheets that connect with a wide basin-covering ice shelf.

## 4.2    Effects of Lomonosov Ridge grounding

The geological data suggests that at least during the MIS 6, the ice shelf grounded on the Lomonosov Ridge (Jakobsson et al., 2016). The transient evolution is dynamically complex when ice shelf begins to ground on the Lomonosov Ridge: There

are now two new moving grounding lines. We do not try to describe the transient dynamics following the grounding event. However, the ice-shelf complex will adjust to a new quasi equilibrium with an ice-surface gradient over the Ridge, which balances the basal drag and drives a mass flux that matches those in the floating ice on either side of the Ridge. After this adjustment, we expect that the ice-shelf thickness on the upstream side in the Amerasian Basin has increased notably, whereas the downstream ice-shelf thickness in the Eurasian Basin has changed only modestly.

In the Eurasian Basin, the ice-shelf mass balance is now given by

$$(x_f - x_r)\frac{\partial h_E}{\partial t} = \int_{x_r}^{x_f} a(x,t)\,dx + q_r(h_A, h_E) - q_f(h_E); \tag{52}$$

and in the Amerasian Basin by

$$(x_r - x_g)\frac{\partial h_A}{\partial t} = \int_{x_g}^{x_r} a(x,t)\,dx + q_g(h_g, h_A) - q_r(h_A, h_E), \tag{53}$$

where $x_r$ is the position of the Lomonosov Ridge.





In a steady state, the ice-shelf thickness can be determined by working upstream from the Fram Strait as follows. By adding the steady-state versions of Eqs. (41,52,53), one finds that the ice-shelf thickness in the Eurasian Basin $h_E$ is determined by

$$q_f(h_E) = \int\limits_{x_0}^{x_f} a(x)\,dx, \tag{54}$$

which essentially identical to Eq. (42) describing the ungrounded case. The ice thickness in the Amerasian Basin is determined by the accumulation upstream of the Ridge from the relation

$$q_r(h_A, h_E) = \int\limits_{x_0}^{x_r} a(x,t)\,dx, \tag{55}$$

which is obtained adding the steady-state versions of Eqs. (41,53). The downstream grounding-line thickness $h_2$ on the Lomonosov Ridge can now be calculated from the mass-flux condition given by Eq. (38). Lastly, the continental grounding-line thickness is found by equating the local ice flux there with $q_g(h_g, h_A)$; see Eqs. (22,41). One noteworthy aspect of the equilibrium dynamics is that the ice thickness in the Eurasian Basin $h_E$ is independent of how surface accumulation and continental ice fluxes to the shelf are distributed. The ice thickness in the Amerasian Basin, on the other hand, depends on how the net accumulation is distributed between the Basins: The ice-shelf thickness in the Amerasian Basin increases with the proportion of the total accumulation supplied to this region if $q_r$ increases with $h_A$.

Figure 8 illustrates qualitatively how the equilibrium ice-shelf thickness changes with the net integrated accumulation, which equals the mass flux. In this illustration, the Fram Strait mass flux is somewhat arbitrarily represented as $q_f(h) \sim h^2$ and it is assumed that the net accumulation is equally divided between the Amerasian and Eurasian parts of the ice shelf, implying that $q_r = q_f/2$. Further, we assume that $h_2 \approx h_E$ and use Eq. (36) to calculate $h_A$. For small values of accumulation/mass-flux, the ice shelf is thin and does not ground on the Lomonosov Ridge. In this regime, there is a single ice-shelf thickness in the Arctic Ocean. By increasing the accumulation the ice-shelf thickness grows and eventually grounds on the Lomonosov Ridge. When this occurs, a new equilibrium state develops that is associated with an ice-thickness difference over the Lomonosov Ridge. As the mass flux is increased further, the ice thickness increases faster in the Amerasian Basin than in the Eurasian one, yielding an increasing hydrostatic pressure gradient in the grounded ice on the Lomonosov Ridge. Note that at the instance of grounding, there are in principle two different ice flow configurations over the Lomonosov Ridge: one where the upstream grounding line advances forward when the mass flux increases and one where it retreats backward to the Amerasian side of the ridge; see Fig. 7. As will be discussed below, the latter solution is presumably unstable.

A fundamental question is whether equilibrium ice-shelf–ice-shelf configuration with partial grounding on the Lomonosov Ridge are stable. To isolate and illuminate the dynamics associated with the ice flux across the Ridge, we consider an idealised case in which the continental ground-line mass flux is taken to be fixed and we ignore possible changes in accumulation rates associated with shifts of grounding-line positions. Moreover, we assume that at time scales over which the ice volumes change in the Amerasian and Eurasian Basins, the rate of change of the grounded ice-volume on the Lomonosov Ridge can be neglected. In this limit, the mass fluxes across the two grounding lines on the ridge are equal and the ridge mass flux takes



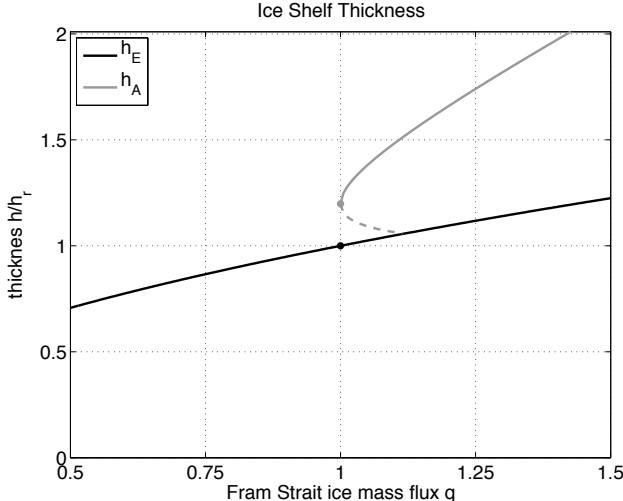

**Figure 8.** A conceptual illustration of how the steady-state ice-shelf thickness varies with the Fram Strait ice export, which equals the net accumulation over the ice shelf. The Amerasian Basin ice-shelf thickness $h_A$ is obtained from Eq. (36). The ice thicknesses are non-dimensionalised with the Lomonosov Ridge crest flotation thickness $h_r$ and the ice export is normalised to be unity when the shelf grounds on the ridge. When $q < 1$, the ice shelf is ungrounded and it has a single thickness over the Arctic Ocean. When the ice shelf grounds on the Ridge, an ice-shelf thickness difference develops between the Amerasian and Eurasian basins. The dotted line depicts a hypothetical solution with both grounding lines on the upstream side of the ridge, which presumably is unstable.

the form given in Eq. (39). Using these assumptions, we follow similar procedures as in section 4.1 and make a linear stability analysis, where the ice thicknesses in the Amerasian and Eurasian Basins are perturbed slightly from their equilibrium values: The linearised version of the mass balance relations (52,53) then become

$$(x_f - x_r)\frac{\partial h'_E}{\partial t} = -\left(\frac{\partial q_f}{\partial h_E} - \frac{\partial q_r}{\partial h_E}\right)h'_E + \left(\frac{\partial q_r}{\partial h_A}\right)h'_A; \tag{56}$$

and

$$(x_r - x_g)\frac{\partial h'_A}{\partial t} = -\left(\frac{\partial q_r}{\partial h_E}\right)h'_E - \left(\frac{\partial q_r}{\partial h_A}\right)h'_A, \tag{57}$$

where $h'_E$ and $h'_A$ are perturbation from an equilibrium state and the partial derivates of the ice mass fluxes are evaluated at the equilibrium state. By making a standard stability analysis of Eqs. (56,57) (Bender and Orzag, 1987), one finds that equilibrium solutions are stable if

$$\frac{\partial q_r}{\partial h_A} > 0, \ \frac{\partial q_r}{\partial h_E} < 0, \ \frac{\partial q_f}{\partial h_E} > 0; \tag{58}$$

This stability criteria is straightforward to interpret physically in the limiting case where the Eurasian ice-shelf segment is small and responds much faster than the Amerasian one: Suppose that the Amerasian ice-shelf thickness has been increased slightly (i.e. $h'_A > 0$) and remains nearly constant as $h'_E$ adjusts. From Eq. (56), we see that a positive $h'_A$ serves to increase





the mass flux over the Ridge into the Eurasian Basin, where the ice thickness begins to grow. If $\partial q_r/\partial h_E < 0$, then the mass flux over the Ridge will decrease with increasing $h'_E$, which represents a stabilising negative feedback. According to the ridge mass-flux relation Eq. (37), this should apply if the downstream grounding line is on the Eurasian side where the bed slopes downward. An additional stabilising feedback comes from the Fram Strait ice export, which increases with the ice thickness in the Eurasian Basin.

If the downstream grounding-line thickness on the Ridge $h_2$ is assumed to be proportional to $h_E$, it follows from Eq. (37) that the equilibrium is stable if the downstream grounding line is on the Eurasian side where the bed slopes downward: Here $\partial q_r/\partial h_A > 0$ and $\partial q_r/\partial h_E < 0$ implying stable equilibrium states. This can be inferred directly from an inspection of Fig. 7b. For the other equilibrium configuration, where the downstream grounding line is on the Amerasian side of the Ridge on an upward-sloping bed, the equilibrium is potentially unstable as $\partial q_r/\partial h_E > 0$: The mass flux over the Ridge increases with increasing Eurasian ice shelf-thickness; see Eq. (37) and Fig. 7a. This represents a positive destabilising feedback. The stability features in this case will be decided of the relative strengths of the positive and the negative feedbacks involved, which depend on the details of $q_r$ and $q_f$ as well as the relative ice-shelf extents $x_r - x_g$ and $x_f - x_r$. However, our analyses of the ice flow over the Lomonosov Ridge and in the Fram Strait are to crude to assess stability features in this case.

In summary, the simple ridge mass-flux formula Eq. (37) combined with a linear stability analysis suggest that the ice-shelf complex should be stable if the downstream grounding line on the Lomonosov Ridge is located on a downward-sloping bed. It is relevant to note that the situation here is different from the classic marine ice sheet stability problem (Weertman, 1974; Schoof, 2007a, 2012): there are two grounding lines and the mass flux at the downstream one decreases rather than increases with local ice thickness. The different relation between grounding-line thickness and mass flux arises because we are considering a non-divergent quasi-equilibrium ice mass flow over the ridge. Thus, the ice flow could be unstable to perturbations of shorter time-scales that are associated with spatial variations in the mass flux over the ridge. However, if the grounded-ice segment on the ridge would become very extensive, we anticipate that the ice dynamics near the downstream grounding line would be no different from that of an ordinary marine ice sheet. In this limit, the criteria of Weertman (1974) will govern the local stability of the downstream grounding line. If the grounded-ice segment short, on the other hand, the presence of an upstream grounding line can potentially affect stability of the ice-sheet–ice shelf complex. It is beyond the scope to further analyse this challenging problem, but we note that the general approach outlined by Schoof (2012) may be used to examine the stability of ice configurations with multiple grounding lines.

# 5 Discussion

## 5.1 Dynamical analogues and differences with the large Antarctic ice shelves

Our theoretical analyses show that a basin-covering Arctic Ocean ice shelf have two regions with distinctly different dynamics: a vast interior region covering the central Arctic Ocean basin and an exit region towards the Fram Strait (Fig. 2). The exit region, which terminates at a calving front towards the Norwegian–Greenland Seas, should have characteristics in terms of extent and dynamics resembling those of present-day large ice shelves around Antarctica. The interior region, which can be viewed as





dammed rather than embayed, should feature a different dynamics in which pronounced back stress supports kilometre thick shelf ice with negligible horizontal thickness variations. The back stress is proportional to mean ice thickness in the interior region and hence increases with increasing ice flow resistance in the exit region.

This conceptual two-regime division of the Arctic Ocean ice shelf is partly motivated by the smallness of the non-dimensional parameter $\epsilon$ [Eq. (15))], which measures the ratio between viscous stresses and the hydrostatic pressure gradient (Schoof, 2007b; Goldberg et al., 2009): When $\epsilon$ is small, the variations in ice-shelf thickness should be small compared with the mean ice thickness. For the large Antarctic ice shelves corresponding values of $\epsilon$ are in fact very small (Goldberg et al., 2009) and qualitative comparable to that characterising the Arctic Ocean ice shelf. Notably, the Ross Ice Shelf has an outer segment extending over 500 km in the stream-wise direction, where the ice-shelf thickness is fairly uniform up to the calving front (MacAyeal and Thomas, 1986). Because the ice shelf terminates at a broad open-ocean calving front, however, buttressing and back stresses are negligible in this outer segment of the Ross Ice Shelf (Thomas and MacAyeal, 1982). In contrast, the interior Arctic Ocean ice shelf meets the Fram Strait constriction, where local boundary stresses create an effective back stress in the upstream ice shelf. The constricted Ross Ice Shelf segment inside Roosevelt Island should have some dynamical similarities with an Arctic Ocean ice shelf. Although this part of the Ross Ice Shelf extends only some 300 km, it features modest ice-thickness variations and significant back stresses (MacAyeal and Thomas, 1986; Thomas and MacAyeal, 1982). Accordingly, it is the specific embaying geometry rather than the size that gives the interior Arctic Ocean ice shelf its distinctive dynamics. In fact, the interior Arctic Ocean ice shelf should have dynamical similarities with globally ocean-covering ice shelves that may have formed during Snow Ball Earth events (Li and Pierrehumbert, 2011): such ice shelves are completely confined horizontally.

## 5.2 Spatial distribution of the glacial ice shelf

The present theoretical considerations and review of recent ice-dynamical studies suggest that the dynamically most consistent scenario is a fully-developed glacial Arctic Ocean ice shelf of approximately uniform thickness that cover the entire basin. This is in line with numerical-modelling results of Colleoni et al. (2016a) and the reconstruction based on mapped ice grounding on submarine ridges and bathymetric highs of Jakobsson et al. (2016) shown in Fig. 1, which in turn is similar to the earlier ice-shelf scenario proposed by Mercer (1970), Hughes et al. (1977) and Grosswald and Hughes (1999). The mapped mega-scale lineation features on the Arlis Plateau and southern Lomonosov Ridge off the New Siberian Islands point to an ice shelf drafting deeper than 1000 m below present sea level, which is incompatible with the notion of a freely spreading ice shelf that only covers fractions of the Arctic Ocean: Without back stresses from ice-shelf embayment towards the Fram Strait, the ice mass flux from the continental ice sheets would have been unreasonably large (Schoof, 2007b). It is important to note that for the same physical reason, a kilometre thick Amerasian Basin ice shelf, which grounds on the Lomonosov Ridge and thereafter spreads freely in the Eurasian Basin is dynamically unfeasible: the mass flux from the downstream grounding line on the Lomonosov Ridge would be gigantic, causing rapid thinning and un-grounding of the ice shelf. Further, our discussions have concerned an ice shelf that terminates near the Fram Strait. It is open question whether glacial Arctic Ocean ice shelves have continued southward to the North Atlantic covering also the Nordic Seas as proposed for instance by Hughes et al. (1977)



and Lindstrom and MacAyeal (1986). Mapped evidence of ice grounding on the Hovgaard Ridge south of the Fram Strait (Arndt et al., 2014) indicate that ice shelves extending into the Norwegian–Greenland Seas cannot not be firmly dismissed at this stage.

### 5.3 The puzzling absence of LGM ice-shelf traces

A noteworthy result of our theoretical analyses is that the equilibrium thickness of an Arctic Ocean ice shelf should be fairly insensitive to the spatial distributions of surface accumulation, basal oceanic melt, and influx of continental ice. Further, if there exists a Fram Strait ice-export relation that depends mainly on the mean Arctic ice-shelf thickness, then the ice-export dynamics should have been similar during the Marine Isotope Stage 6 and the Last Glacial Maximum. If these assumptions hold, the equilibrium ice-shelf thicknesses at these two glacial stages should depend primarily on the total mass supply to the ice shelf.

In the Arctic Ocean, essentially no documented ice-erosion traces below depths of around 600 m have been dated to the LGM (Jakobsson et al., 2014). Taken at face value, this implies that if an Arctic Ocean ice shelf was formed during the LGM its thickness was less than about 500 m, which is roughly half of the inferred MIS 6 thickness. By assuming equilibrium ice-shelf conditions and a knowledge of the Fram Strait ice-export relation, one could estimate the difference in net accumulation between these two glacial stages. If the ice export is proportional to the ice-shelf thickness, the net accumulation should have been about twice as large during the MIS 6 than during the LGM. On the other hand, if the ice export increases faster than linearly with the ice-shelf thickness, then a larger net accumulation difference is needed to yield a factor of two in ice-shelf thickness difference. The hypothetical ice-export relation given by Eq. (25), which depends on the fourth power of the thickness, requires a 16 fold increase in accumulation to give a doubled equilibrium ice-shelf thickness. Thus, the sensitivity of the Fram Strait ice export may determine if a difference in net accumulation can be a plausible explanation for a significantly thinner ice shelf during the LGM.

Climate-modelling studies give some support to the notion that the mass supply to an Arctic Ocean ice shelf could have been higher during the glacial conditions of MIS 6. A main factor is that the Laurentide Ice Sheet presumably was lower and smaller than during the LGM (Colleoni et al., 2016b). Studies with atmospheric-circulation models show that a high Laurentide Ice Sheet tend to make the Atlantic jet stream more zonal, which shifts the cyclone paths southward over Europe; whereas a low Laurentide Ice Sheet yields a more modern-day North Atlantic atmospheric circulation with cyclone paths reaching northwestern Europe and the Arctic (Löfverström et al., 2014; Colleoni et al., 2016b). Thus, a smaller MIS 6 Laurentide Ice Sheet may have resulted in an atmospheric-circulation regime that was more conducive for building a massive Artic Ocean ice shelf. Modelling experiments of Colleoni et al. (2016a) suggest that compared to the LGM, surface accumulation rates over the Arctic Ocean may have been some 50 % higher during the MIS 6.

Oceanic basal melting is a leading-order component in the mass balance of present-day Antarctic ice shelves (Jacobs et al., 2011; Rignot et al., 2013). Moreover, numerical ice-shelf–ice-sheet modelling suggests that variations oceanic basal melting have played a crucial role for collapses and advances of the West Antarctic ice sheet during the past 5 million years (Pollard and DeConto, 2009). Basal melting is also expected to be important for an Arctic Ocean ice shelf. Building on a Ross Ice Shelf





study by MacAyeal (1984), Jakobsson et al. (2016) developed a conceptual oceanographic model of the circulation in a huge flat-roofed ice cavity, representative of an Arctic Ocean ice shelf. In their model, plausible variations of the weakly constrained parameters such as oceanic temperature and vertical mixing can give changes in basal melting that have a larger impact on the ice-shelf mass balance than the ones who Colleoni et al. (2016a) attributed to atmospheric-circulation differences between

the MIS 6 and the LGM. Marine sedimentary records suggest that immediately before and during the LGM, the deep Arctic Ocean and Nordic Seas were several degrees warmer than today (Cronin et al., 2012; Thornalley et al., 2015). Despite that no comparable information exits for the MIS 6, this raises the possibility that differences in oceanic conditions may explain the absence of a thick LGM ice shelf.

Can the time available to build ice shelves explain the difference between the LGM and the MIS 6? Analyses of stable

isotope records indicate that MIS 6 represents a longer period of fully-glaciated conditions that the LGM (Jouzel et al., 2007). With plausible net accumulation rates in the range of 0.05–0.1 m year$^{-1}$ (Bigg and Wadley, 2001; Colleoni et al., 2016a) and in the absence of ice export, it would take some 10 000 to 20 000 years to build a kilometre thick ice shelf[6]. Evidently, Fram Strait ice export slows the ice-shelf growth. By how much will depend on the relation between ice export and ice-shelf thickness. If the export increases sharply with ice thickness, the export loss becomes significant in the mass balance first when the ice-shelf

thickness approaches the equilibrium thickness. In this case, the ice shelf reaches its equilibrium thickness at a time essentially set by the net accumulation rate. A weaker sensitivity of the ice export on ice thickness can significantly prolong the time required to reach the equilibrium ice-shelf thickness.

## 5.4   Threshold effects and Arctic ice-sheet stability

Hughes et al. (1977) emphasised the stabilising role of a glacial Arctic Ocean ice shelf for the contiguous marine ice sheets,

and proposed a scenario for how ice-shelf disintegration could lead to rapid de-glaciation. Such "uncorking" effects of marine ice sheets, associated with rapid loss of buttressing, may occur when the ice-shelf thickness passes certain thresholds controlled by bed and basin geometries (Schoof, 2007a; Pollard and DeConto, 2009). A critical stability point for Arctic marine sheets may be passed when a glacial ice shelf un-grounds from the Lomonosov Ridge: Figure 8 indicates that the Amerasian Basin ice-shelf thickness may change sharply when the ice shelf grounds on or un-grounds from the Lomonosov Ridge. Moreover,

when the ice shelf becomes afloat from the Lomonosov Ridge, oceanic heat transport into the Amerasian Basin ice cavity is expected to increase, which could further erode the ice shelf. A rapid decrease in ice-shelf thickness reduces drastically the buttressing of the contiguous ice sheets (Hughes et al., 1977; Dupont and Alley, 2005; Goldberg et al., 2009). As a response, the grounding-line stress and mass flux increase, causing the continental ice sheet to surge and the grounding line to retreat. The increased ice-sheet mass flux to the Arctic Ocean may cause the ice shelf to temporarily re-ground on the Lomonosov

Ridge, yielding an oscillatory adjustment towards a new equilibrium. Accordingly, changes in climatic conditions that cause the ice shelf to ground and and un-ground on the Lomonosov Ridge may trigger ice-sheet surges to the Arctic Ocean, which

---

[6]In 30 000 model-year long numerical simulations, Colleoni et al. (2016a) have obtained basin-covering Arctic Ocean ice shelves with thicknesses reaching over 2 km. However, details of ice-thickness evolution and Fram Strait ice export are nor provided in the paper.




in some respects could resemble Heinrich events in the North Atlantic Ocean (Clarke et al., 1999), or surges associated with instabilities of the West Antarctic ice sheet (Pollard and DeConto, 2009; Joughin and Alley, 2011).

The Yermak Plateau at the northern edge of the Fram Strait (Fig. 1) could also serve as a dynamical switch of the ice shelf. This is a broad and heavily ice eroded plateau with a depth of around 600 m below the present sea level (Dowdeswell et al., 2010; Jakobsson et al., 2014). When an ice shelf becomes thick enough to ground on the Yermak Plateau, basal drag should increase sharply and ice export through the Fram Strait should decrease temporarily. To reattain equilibrium conditions, in which ice export and net accumulation balance, the Arctic Ocean ice-shelf thickness needs to increase. Thus, ice-shelf grounding on the Yermak Plateau could represent another threshold, associated with a sharp increase in equilibrium ice-shelf thickness, which could have been passed during the MIS 6 but not during the LGM.

## 5.5 Concluding remarks

Motivated by earlier hypotheses of Arctic Ocean ice shelves during glacial conditions (Mercer, 1970; Hughes et al., 1977; Grosswald and Hughes, 1999) and new observationally-based evidence (Dowdeswell et al., 2010; Jakobsson et al., 2010; Niessen et al., 2013; Jakobsson et al., 2016), we have theoretically analysed the physical features of an extensive and effectively dammed Arctic Ocean ice shelf. The present results indicate that some puzzling glacial questions concerning formation of kilometre-thick basin-covering ice shelves hinges on the ice-shelf flow resistance through the Fram Strait and its associated ice-shelf back stresses in the central Arctic Ocean. Appropriately designed simulations with numerical ice-shelf models should be able to provide critical information on this issue. Furthermore, numerical ice-shelf modelling may be used to analyse how the two-dimensional Arctic Ocean ice-flow pattern depends on the distribution of ice flux from the Arctic continental margins and local surface mass balance. Combined with the data-inferred ice flow directions, such numerical experiments could provide important constraints on ice-shelf features and continental ice sheet configurations: the hypothesised existence of an East Siberian ice sheet during MIS 6 has been difficult to confirm on the basis of terrestrial records (Niessen et al., 2013; Jakobsson et al., 2014). Another intriguing question is how ocean–ice interactions operate in the special stetting of an Arctic Ocean ice shelf. Although studies on Antarctic ice shelves provide useful insights (MacAyeal, 1984; Jacobs et al., 2011; Rignot et al., 2013), the glacial Arctic Ocean should entail some different aspects of ocean–ice interactions. Distinguishing features of the glacial Arctic Ocean include very weak tidal currents and a vast flat-roofed ice cavity with water depths exceeding 2 km, which has a highly isolated "Snow Ball Earth" like Amerasian Basin compartment (Ashkenazy et al., 2013). In conclusion, glacial Arctic Ocean ice shelves feature a rich and interesting set of dynamical questions that can provide illuminating perspectives on the dynamics and evolution of Antarctic ice shelves as well as analogies with glacial Snow Ball Earth scenarios.

## 6 Code availability

Not applicable.



## 7 Data availability

Not applicable.

*Author contributions.* Discussions between JN and MJ initiated the work. JN has been leading the writing of the paper receiving critical input from all the authors.

5 *Competing interests.* N/A

*Disclaimer.* N/A

*Acknowledgements.* This research is a part of the SWERUS-C3 project financed by Knut and Alice Wallenberg Foundation, Swedish Polar Research Secretariat and Stockholm University. Research grants to individual scientist were provided by the Swedish Research Council and the Swedish National Space Board. We thank Jonas Nycander for interesting discussions on ice dynamics.





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
