# Peer review of "Ice-shelf damming in the glacial Arctic Ocean: dynamical regimes of a basin-covering kilometre thick ice shelf"

_The Cryosphere, 2017_

## Referee Comment (RC1) · T. Cronin (Referee) · 9 May 2017

Nilsson-2017-Cryosphere-Review

Main points This is a very well-written manuscript making an important advance on understanding Arctic [and indirectly Antarctic] ice sheet growth and decay. The topic of Arctic-wide ice shelves is receiving much attention due to new discoveries from various expeditions. The ages of past ice shelves is however debated.

The text is long and highly technical, aimed at glaciologists [which I am not], but highly relevant to glacial geologists and paleoclimatologists [which I am]. Needless to say the "Mediterranean" configuration of the Arctic Ocean, with the narrow Fram Strait connection to the Nordic Seas, make is a very unique system for ice shelf behavior. The Discussion section comparing the Ross Ice Shelf to the Arctic Ice Shelf seems especially important for the future of the former, and the history of the latter. So is the unresolved question of MIS 6 versus LGM MIS 2 Arctic Ice shelves and the larger extent of the former. Conceptually Snowball Earth reconstructions are interesting but gee, that was a different world from the Quaternary and much less empirical data on the actual extent of glaciations.

The Mercer-Hughes-Grosswald studies were made in the context of CLIMAP reconstructions and the large vs small LGM ice sheet debates. Questions: What are the implications for global sea level as many authors using stable oxygen isotopes plot LMG sea level lower than that during MIS 6? What are the implications if any for the calibration of deep sea [and Red Sea] oxygen isotopic records, used to address so many other paleoclimate topics, to sea level – any thoughts on what such ice shelves might have on global ocean O18 ratios? Do the glaciological constraints described here reconcile issues in global se level [matching marine terraces to O18 to ice sheet glacial geology]?

Minor points

Throughout: Should kilometer-thick ice sheet have a hyphen ? Also be consistent citing equations in the text Abst line 5 has not have Line 16 page 38 "is assumed to be..." Figure 1 caption & text has lower case a, b, c but the figure itself has upper case A, B, C Page 6 line 21 "further HINDER..." p. 12 line 24 it is worth noting... Page 18 line 16. Something is missing here Page 25 line 30 ice shelf HAS [not have?]

---

## Referee Comment (RC2) · Anonymous Referee #2 · 8 Jun 2017

The paper presents a theoretical analysis of the feasibility and dynamics of an Arctic Ice Shelf during MIS6 and the LGM. The topic in itself is interesting, providing a new perspective on an element which is not fully understood in the reconstruction of past ice sheets in the Northern Hemisphere.

The paper is well presented, however, as Reviewer 1 states, the manuscript is highly technical. The discussion does serve to help a non-mathematician to understand the outcome of the work, but the model description and theoretical analysis in sections 3 & 4 are otherwise somewhat impenetrable to a non-mathematician. By its nature it is a theoretical analysis, so needs to be presented as it is, I wonder if it would be better

submitted to a more mathematical journal, but that it not to say that I do not consider it suitable for the Cryosphere.

I am afraid I cannot comment in depth about the mathematical theory, to my understanding it looks to be relying on all the appropriate sources, but I suggest that it needs a close look over by one of the many authors cited in the model description section to be sure that the conclusions that are drawn from the theoretical analysis are robust.

Corrections:

p1, line 5: should be "the ice shelf has . . ."

p1, line 8: should be "A narrow transition zone. . ."

p2, line 30: should be "These analyses. . ."

p2, line 30: should be ". . . Arctic ice shelf. . ."

p2, line 32: insert ". . . thick enough to ground and erode. . ."

p4, line 11: should be ". . . reaching more than 1000 m . . ."

p4, line 13: ". . .there are few" – should this be "few" or "a few" (very different meaning!)?

p5, line 3: should be ". . .simulated to lie in the range of. . ."

p13, line 13: should this be "amenable" rather than "amendable"?

p29, line 22: should be ". . .special setting . . ."

---

## Editor Comment (EC1) · C. R. Stokes (Editor) · 12 Jun 2017

Dear Johan,

You will, by now, have received formal notification that the open discussion of your manuscript has closed.

I would like to thank Thomas Cronin and the anonymous referee for their comments, both of which were very favourable and were restricted to minor edits. My own reading of the manuscript is that it represents a very valuable contribution to the ongoing debates about the viability and timing of Arctic Ocean ice shelves. As the reviewers note,

some aspects are highly technical, but these sections are required and this manuscript is particularly impressive in terms of explaining and discussing the key results and implications of the modelling. As such, I would certainly encourage you to submit a revised manuscript.

In addition to the reviewer edits, do carefully check your use of commas versus full-stops when referring to dates (e.g. line 2 of page 2 – the latter should be a full stop). In relation to section 5.2, it would also be helpful to quantify what you mean by "gigantic" in relation to the mass flux.

I look forward to receiving a revised version of the manuscript in due course.

If you have any queries, please do not hesitate to get in touch.

Kind regards,

Chris Stokes Editor

---

## Author Response (AR1)

Dear Chris Stokes (editor),

we have now revised our manuscript based on the comments from the reviewers and the editor. The response to the reviewers is detailed below. Regarding the phrasing "gigantic" in section 5.2, we have made a calculation based on the mass flux formula Eq. (22) – as described in a new footnote – and re-phrased "gigantic" to "very large", which is more appropriate. We have checked for the usage of commas versus full-stops when referring to dates, and made changes. We have also corrected a few additional typos and added a few new references when addressing the reviewers' comments.

Best regards,
Johan Nilsson

**Response to reviewer 1**

We appreciate the positive and constructive review comments provided by Dr. Thomas Cronin. The significance of extensive Arctic Ocean ice shelves for the translations of oceanic oxygen isotopic records into glacial–interglacial sea-level changes is a relevant and interesting point. We deem that it is beyond the scope of the present paper to give a detailed account on existing estimates of the sea level during the LGM and the MIS 6. As the reviewer suggests, however, it is relevant to mention the role of Arctic Ocean ice shelves in connection with glacial sea levels. For this purpose, we will in the revision add the following paragraph at the end of section 5.3:

"A significant difference in volume of the Arctic ice shelves during the LGM and the MIS 6 has implications when inferring glacial sea levels from ocean oxygen isotope records. Broecker (1975) emphasised that floating ice shelves have essentially no impact on the sea level, but that the isotopically-light oxygen contained in ice shelves increases the mean oceanic $\delta^{18}O$ value. Thus, temperature-corrected $\delta^{18}O$ data from the deep ocean as well as from the Red Sea will overestimate glacial sea-level decreases if large ice shelves were present. The fully-developed MIS 6 Arctic Ocean ice shelf should have increased the mean oceanic $\delta^{18}O$ value by roughly 0.15 permil, equivalent to the increase caused by a sea-level decrease of around 15 meters due to continental ice accumulation (Mix and Ruddiman, 1984; Jakobsson et al., 2016). It can be noted that the deep ocean $\delta^{18}O$ values were similar during MIS 6 and LGM (Lisiecki and Raymo, 2005). By accounting for the difference in ice-shelf volume, the deep ocean $\delta^{18}O$ record indicates that the sea level should have been some 5 to 10 meters lower during LGM compared to MIS 6. However, it may be challenging

to infer volumes of glacial Arctic ice shelves from comparisons of sea level estimates based on landforms and oxygen isotopes. The reason is that the oceanic $\delta^{18}$O signature of Arctic ice shelves is comparable to uncertainties related to various steps in deriving $\delta^{18}$O values and assumptions involved in converting them to sea level (Siddall et al., 2004). Still, it may be worthwhile to revisit the question of the relative sea levels during MIS 6 and LGM in light of the new data-based support of an extensive MIS 6 ice shelf in the Arctic Ocean."

We have also address the minor issues related to spelling and typos in the revision.

**Response to reviewer 2**

We have corrected the typos pointed out by the reviewer.

[revised manuscript text omitted]